# Hierarchical Integration Diffusion Model for Realistic Image Deblurring

**Zheng Chen**[1], **Yulun Zhang**[2*], **Ding Liu**[3], **Bin Xia**[4], **Jinjin Gu**[5,6], **Linghe Kong**[1*], **Xin Yuan**[7]

[1]Shanghai Jiao Tong University, [2]ETH Zürich, [3]Bytedance Inc, [4]Tsinghua University,
[5]Shanghai AI Laboratory, [6]The University of Sydney, [7]Westlake University

## Abstract

Diffusion models (DMs) have recently been introduced in image deblurring and exhibited promising performance, particularly in terms of details reconstruction. However, the diffusion model requires a large number of inference iterations to recover the clean image from pure Gaussian noise, which consumes massive computational resources. Moreover, the distribution synthesized by the diffusion model is often misaligned with the target results, leading to restrictions in distortion-based metrics. To address the above issues, we propose the Hierarchical Integration Diffusion Model (HI-Diff), for realistic image deblurring. Specifically, we perform the DM in a highly compacted latent space to generate the prior feature for the deblurring process. The deblurring process is implemented by a regression-based method to obtain better distortion accuracy. Meanwhile, the highly compact latent space ensures the efficiency of the DM. Furthermore, we design the hierarchical integration module to fuse the prior into the regression-based model from multiple scales, enabling better generalization in complex blurry scenarios. Comprehensive experiments on synthetic and real-world blur datasets demonstrate that our HI-Diff outperforms state-of-the-art methods. Code and trained models are available at https://github.com/zhengchen1999/HI-Diff.

## 1 Introduction

Image deblurring is a long-standing computer vision task, aiming to recover a sharp image from a blurred observation. Blurred artifacts can arise from various factors, *e.g.*, camera shake and fast-moving objects. To address this challenge, traditional methods typically formulate the task as an optimization problem and incorporate natural priors [12, 38, 24, 29] to regularize the solution space. However, since blur in real scenarios is complex and non-uniform, which is hard to be modeled by the specific priors, these algorithms suffer from poor generalization in complex situations.

With the development of deep learning, convolutional neural networks (CNNs) have been applied in image deblurring [37, 43, 13, 55, 1, 11]. Among them, the regression-based methods have shown remarkable success, especially in terms of distortion-based metrics (*e.g.*, PSNR). Moreover, Transformer-based approaches [48, 53, 44, 21], which can capture long-distance dependencies, are introduced as an alternative to CNNs. These methods further boost the deblurring performance. However, regression-based methods are prone to recovering images with fewer details, since the regression losses are conservative with high-frequency details [36].

Apart from regression-based methods, deep generative models, like generative adversarial networks (GANs) [14] and normalizing flows [8], provide other solutions for generating complex details. Recently, Diffusion Models (DMs) [40, 15] have exhibited impressive performance in image synthesis [41, 35] and restoration tasks (including image deblurring) [50, 33, 36, 51]. DMs generate high-fidelity images through a stochastic iterative denoising process from a pure white Gaussian noise. Compared to other generative models, such as GANs, DMs generate a more accurate target distribution without encountering optimization instability or mode collapse.

---

*Corresponding authors: Yulun Zhang, yulun100@gmail.com; Linghe Kong, linghe.kong@sjtu.edu.cn

37th Conference on Neural Information Processing Systems (NeurIPS 2023).

Nonetheless, DM-based methods face several challenges. **First**, DMs are limited by the high computational cost of generating samples. It requires a high number of inference steps. Some methods reduce the number of iterations by predicting the residual distribution [50] or applying advanced sampling strategies [41, 3]. However, the overall computational complexity is still high, especially for high-resolution images. Furthermore, although some methods perform DMs on latent space [35], the compression ratio is small (*e.g.*, 8 times), due to limitations in generation quality. **Second**, generative models, including DMs, tend to produce undesired artifacts not present in the original clean image. Besides, the generated details may also be misaligned with real targets. These lead to the poor performance of DM regarding some distortion-based metrics (*e.g.*, PSNR).

The aforementioned issues promote us to apply DMs from another perspective. The motivation for our work is threefold. **First**, due to the high computational overhead in the image space, we also consider applying DMs on the low-dimensional latent space. Meanwhile, we increase the compression ratio of latent to effectively reduce the computational complexity. **Second**, since the advantages of regression-based methods in distortion accuracy, we integrate DMs and regression-based methods to improve the performance of DMs regarding distortion. **Third**, considering the non-uniform blur in real scenarios, we apply a hierarchical approach to enhance the generalization of the method.

Specifically, we apply DMs to generate priors in latent space, inspired by the application of priors in traditional deblurring algorithms. The priors are integrated into the regression-based model in a hierarchical manner to improve the details of restored images. The design contains three benefits: **(1)** The dimension of latent space can be very low, since the regression-based model restores most of the distribution. **(2)** The issue of distortion caused by misaligned details generated by DMs can be avoided. **(3)** The hierarchical integration enables better generalization in complex blurry scenarios.

Based on the above analysis, we propose a novel approach called the Hierarchical Integration Diffusion Model (HI-Diff) for realistic image deblurring. Following previous practice [10, 35, 51], we perform a two-stage training to realize latent compression and the training of the DM, respectively. Without loss of generality, for the regression-based methods, we choose the encoder-decoder Transformer architecture. **In the first stage**, we compress the ground-truth image into a highly compact latent representation as the prior feature through a latent encoder (LE). We propose the hierarchical integration module (HIM) that can fuse the prior and intermediate features of Transformer at multiple levels. We jointly train the LE and Transformer to effectively construct the prior. **In the second stage**, we train a latent diffusion model to generate the prior feature in the latent space from the Gaussian noise and guide Transformer through the HIM. Similar to the first stage, we jointly train the DM and Transformer. In summary, our main contributions are three-fold as follows:

- We propose the Hierarchical Integration Diffusion Model (HI-Diff), for realistic image deblurring. The HI-Diff leverages the power of diffusion models to generate informative priors, that are integrated into the deblurring process hierarchically for better results.

- We apply the diffusion model in highly compact latent space to generate the prior. Meanwhile, we propose the hierarchical integration module to fuse the prior into the regression-based model from multiple scales, enabling the generalization in complex blurry scenarios.

- Extensive experiments conducted on synthetic and real-world blur datasets demonstrate the superior performance of the HI-Diff in comparison to state-of-the-art deblurring methods.

## 2 Related Work

### 2.1 Image Deblurring

**Traditional Methods.** Traditional deblurring methods generally formulate the problem as an optimization problem [12, 38, 24, 6, 29]. They utilize various natural image priors for sharp images and blur kernels. Common methods include local smoothness prior [38], sparse image prior [24], $L_0$-norm gradient prior [52], and dark channel prior [29]. However, these methods rely on manually crafted priors, which results in poor generalization ability and limited performance in complex situations.

**Deep CNN-based Methods.** With the rapid development of deep learning, significant progress has been made in image deblurring using CNN-based methods [37, 28, 43, 55, 1, 16]. For instance, MSCNN [28] designs a multi-scale CNN to restore sharp images. SRN [43] proposes a coarse-to-fine scale-recurrent network for more efficient multi-scale image deblurring. DMPHN [55] devises a deep multi-patch hierarchical deblurring network to exploit the deblurring information at different scales. XYDeblur [16], on the other hand, divides the original deblurring problem into two sub-problems.

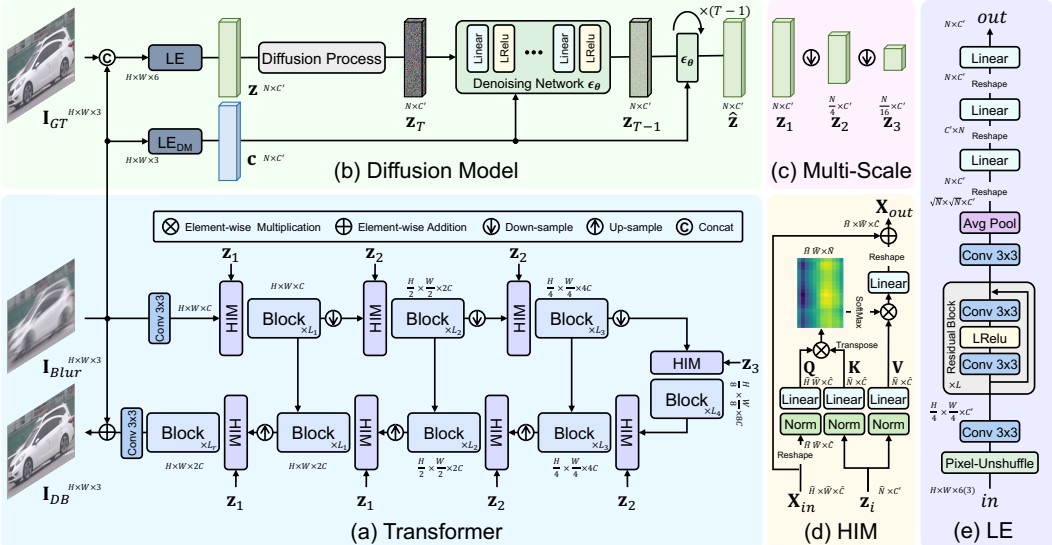

Figure 1: The overall framework of our HI-Diff. (a) Transformer, adopts hierarchical encoder-decoder architecture, equipped with HIM, for the deblurring process. (b) Diffusion Model, is performed in a highly compact latent space for computational efficiency. (c) The multi-scale prior feature $\{\mathbf{z}_1, \mathbf{z}_2, \mathbf{z}_3\}$ is obtained by downsampling the prior feature multiple times. In stage one, $\mathbf{z}_1 = \mathbf{z}$; in stage two, $\mathbf{z}_1 = \hat{\mathbf{z}}$. (d) The hierarchical integration module (HIM), calculates cross-attention between the intermediate feature of Transformer and the multi-scale prior feature. (e) The latent encoder (LE), where the size of the input feature ($in$) is $H \times W \times 6$ for LE, and $H \times W \times 3$ for $\text{LE}_{\text{DM}}$.

**Transformer-based Methods.** Recently, Transformer [46, 9] proposed in natural language processing (NLP), which utilizes the self-attention (SA) mechanism to model long-range dependence, has shown remarkable performance in image deblurring [48, 53, 44]. For example, Restormer [53] utilizes "transposed" attention, which calculates the SA across channel dimension instead of spatial dimension, to model global information and keep computational efficiency to large images. Stripformer [44] constructs horizontal and vertical strip tokens to catch region-specific blurred patterns with different orientations in dynamic scenes. These methods further improved the deblurring performance compared with CNN-based Methods. However, they are limited in recovering image details, since the regression-based methods are conservative with high-frequency details [36].

## 2.2 Diffusion Models

Diffusion Models (DMs) [40, 15] are probabilistic generative models that enable the construction of desired data samples from the Gaussian noise via a stochastic iterative denoising process. DMs have demonstrated excellent performance in various image restoration tasks [18, 17, 47, 7], such as super-resolution [36, 25], inpainting [35], and deblurring [50]. A survey [26] summarizes more diffusion-based restoration methods. For example, DiffIR [51] utilizes the diffusion model to generate a prior representation for image restoration, and applies a two-stage training approach. DvSR [50] introduces conditional DMs into image deblurring with a residual model, showing more realistic results than regression-based methods. InDI [7] directly produces high-quality results by restoring low-quality images. However, one of the restrictions of DMs is the large number of iteration steps required in inference. Some methods attempt to mitigate this limitation by applying well-designed noise schedules [3] and sampling strategies [41] or performing DM on the latent space [35]. Despite these efforts, the overall complexity remains high, especially for high-resolution images common in image deblurring. Additionally, these methods are susceptible to misaligned details distribution and undesired artifacts, resulting in poor performance in distortion-based metrics, *e.g.*, PSNR.

## 3 Method

Our goal is to integrate the diffusion model (DM) and the regression-based model, thus overcoming the shortcomings of the DM and realizing better deblurring. We propose the Hierarchical Integration Diffusion Model (HI-Diff). The HI-Diff performs the DM to generate the prior feature that is integrated hierarchically into Transformer (the regression-based model).

The overall framework of the proposed HI-Diff is depicted in Fig. 1. The HI-Diff consists of two parts: Transformer and the latent diffusion model. We adopt Restormer [53], a hierarchical encoder-decoder Transformer architecture, in our method. Compared with other Transformer networks designed specifically for image deblurring, Restormer is a general restoration model. Applying this model in our method can better illustrate the effectiveness of our proposed method. Meanwhile, following previous practice [10, 35, 51], we train our HI-Diff with a two-stage training strategy, to realize latent compression and the training of the DM. In this section, we first elaborate on the two-stage training framework and then illustrate the whole deblurring inference process.

## 3.1 Stage One: Latent Compression

In stage one, our purpose is to compress the ground truth image into the highly compact latent space, and utilize it to guide Transformer in the deblurring process. As shown in Fig. 1(a, b), we compress the ground truth images through a latent encoder (LE) to obtain a compact representation as the prior feature. Then we integrate the prior feature into Transformer through the hierarchical integration module (HIM). The prior feature can provide explicit guidance for Transformer, thus increasing the details of the reconstructed image. Next, we describe these parts in detail.

**Latent Encoder.** As shown in Fig. 1(b), given the blurry input image $\mathbf{I}_{Blur}\in\mathbb{R}^{H\times W\times 3}$ and its corresponding ground truth counterpart $\mathbf{I}_{GT}\in\mathbb{R}^{H\times W\times 3}$, we first concatenate them along the channel dimension and feed them into the latent encoder (LE) to generate the prior feature $\mathbf{z}\in\mathbb{R}^{N\times C'}$. Here $H$ and $W$ represent the image height and width, while $N$ and $C'$ are the token number and channel dimensions of $\mathbf{z}$. Importantly, the token number $N$ is a constant much smaller than $H\times W$. The compression ratio ($\frac{H\times W}{N}$) is much higher than that applied in previous latent diffusion [35] (*e.g.*, 8 times). Therefore, the computational burden of the subsequent latent diffusion model is effectively reduced. The details of the latent encoder are depicted in Fig. 1(e), which contains $L$ residual blocks.

**Hierarchical Integration Module.** To effectively integrate the prior feature and intermediate feature of Transformer, we propose the hierarchical integration module (HIM). As illustrated in Fig. 1(a), the HIM is placed in front of each encoder and decoder. For each HIM, cross-attention is computed between the prior and intermediate features for feature fusion. This module allows the information in the prior feature to be aggregated into features of Transformer.

Specifically, as shown in Fig. 1(d), given the intermediate feature $\mathbf{X}_{in}\in\mathbb{R}^{\hat{H}\times\hat{W}\times\hat{C}}$, we reshaped it as tokens $\mathbf{X}_r\in\mathbb{R}^{\hat{H}\hat{W}\times\hat{C}}$; where $\hat{H}\times\hat{W}$ is spatial resolution, and $\hat{C}$ denotes channel dimension. Then we linearly project $\mathbf{X}_r$ into $\mathbf{Q}\in\mathbb{R}^{\hat{H}\hat{W}\times\hat{C}}$ (*query*). Similarly, we project the prior feature $\mathbf{z}_i\in\mathbb{R}^{\hat{N}\times C'}$ as $\mathbf{K}\in\mathbb{R}^{\hat{N}\times\hat{C}}$ (*key*) and $\mathbf{V}\in\mathbb{R}^{\hat{N}\times\hat{C}}$ (*value*). The cross-attention is formulated as:

$$\mathbf{Q} = \mathbf{W}_Q\mathbf{X}_r, \mathbf{K} = \mathbf{W}_K\mathbf{z}_i, \mathbf{V} = \mathbf{W}_V\mathbf{z}_i,$$
$$\text{Attention}(\mathbf{Q}, \mathbf{K}, \mathbf{V}) = \text{SoftMax}(\mathbf{Q}\mathbf{K}^T/\sqrt{\hat{C}}) \cdot \mathbf{V}, \qquad (1)$$

where $\mathbf{W}_Q\in\mathbb{R}^{\hat{C}\times\hat{C}}$, $\mathbf{W}_K\in\mathbb{R}^{C'\times\hat{C}}$, and $\mathbf{W}_V\in\mathbb{R}^{C'\times\hat{C}}$ represent learnable parameters of linear projections without bias. As vanilla multi-head self-attention [46, 9], we separate channels into multiple "heads" and calculate the attention operations. Note that Fig. 1(d) depicts the situation with a single head and omits some details for simplification. Finally, we reshape and project the output of cross-attention, and add it with $\mathbf{X}_{in}$ to derive the output feature $\mathbf{X}_{out}\in\mathbb{R}^{\hat{H}\times\hat{W}\times\hat{C}}$.

Moreover, since the non-uniform blur in real scenarios, the single-scale prior feature cannot adapt well to complex blurry situations. Therefore, we generate the multiple-scale prior feature $\{\mathbf{z}_1, \mathbf{z}_2, \mathbf{z}_3\}$ (where $\mathbf{z}_1=\mathbf{z}$), by downsampling the prior feature $\mathbf{z}$, as shown in Fig. 1(c). The multiple-scale prior feature adapts to different scale intermediate features for better fusion. The effectiveness of the hierarchical integration with the multiple-scale prior feature is demonstrated in Sec. 4.2.

**Training Strategy.** To ensure the effectiveness of the latent encoder (LE) in constructing the prior feature, we optimize it jointly with Transformer using the $L_1$ loss function, defined as:

$$\mathcal{L}_{\text{deblur}} = \|\mathbf{I}_{DB} - \mathbf{I}_{GT}\|_1, \qquad (2)$$

where $\mathbf{I}_{DB}$ is the deblurred image, and $\mathbf{I}_{GT}$ represents its corresponding ground truth.

## 3.2 Stage Two: Latent Diffusion Model

In stage two, a latent diffusion model (DM) is trained to learn to generate the prior feature, that enhances the deblurring process of Transformer through HIM.

**Diffusion Model.** Specifically, our latent diffusion model is based on conditional denoising diffusion probabilistic models [15, 3, 36, 35]. The diffusion model involves a forward diffusion process and a reverse denoising process, as illustrated in Fig. 1(b).

In the ***diffusion process***, given a ground truth image, we first adopt the latent encoder (LE) trained in stage one to generate the corresponding prior feature $\mathbf{z} \in \mathbb{R}^{N \times C'}$. We take $\mathbf{z}$ as the starting point of the forward Markov process, and gradually add Gaussian noise to it over $T$ iterations as follows:

$$q(\mathbf{z}_{1:T} \mid \mathbf{z}_0) = \prod_{t=1}^{T} q(\mathbf{z}_t \mid \mathbf{z}_{t-1}), \quad q(\mathbf{z}_t \mid \mathbf{z}_{t-1}) = \mathcal{N}(\mathbf{z}_t; \sqrt{1-\beta_t}\mathbf{z}_{t-1}, \beta_t \mathbf{I}), \tag{3}$$

where $t=1,\ldots,T$; $\mathbf{z}_t$ represents the noisy features at the $t$-th step; $\mathbf{z}_0{=}\mathbf{z}$ for unification; $\beta_{1:T} \in (0,1)$ are hyperparameters that control the variance of the noise; $\mathcal{N}$ denotes the Gaussian distribution. Through iterative derivation with reparameterization [20], Eq. (3) can be written as:

$$q(\mathbf{z}_t \mid \mathbf{z}_0) = \mathcal{N}(\mathbf{z}_t; \sqrt{\bar{\alpha}_t}\mathbf{z}_0, (1-\bar{\alpha}_t)\mathbf{I}), \quad \alpha = 1 - \beta_t, \quad \bar{\alpha}_t = \prod_{i=1}^{t} \alpha_i. \tag{4}$$

In the ***reverse process***, we aim to generate the prior feature from a pure Gaussian distribution. The reverse process is a $T$-step Markov chain that runs backwards from $\mathbf{z}_T$ to $\mathbf{z}_0$. Specifically, for the reverse step from $\mathbf{z}_t$ to $\mathbf{z}_{t-1}$, we use the posterior distribution as:

$$q(\mathbf{z}_{t-1} \mid \mathbf{z}_t, \mathbf{z}_0) = \mathcal{N}(\mathbf{z}_{t-1}; \boldsymbol{\mu}_t(\mathbf{z}_t, \mathbf{z}_0), \frac{1-\bar{\alpha}_{t-1}}{1-\bar{\alpha}_t}\beta_t \mathbf{I}), \ \boldsymbol{\mu}_t(\mathbf{z}_t, \mathbf{z}_0) = \frac{1}{\sqrt{\alpha_t}}(\mathbf{z}_t - \frac{1-\alpha_t}{\sqrt{1-\bar{\alpha}_t}}\boldsymbol{\epsilon}), \tag{5}$$

where $\boldsymbol{\epsilon}$ represents the noise in $\mathbf{z}_t$, and is the only uncertain variable. Following previous work [15, 36, 35, 51], we adopt a neural network (denoted as denoising network, $\boldsymbol{\epsilon}_\theta$) to estimate the noise $\boldsymbol{\epsilon}$ for each step. Since DM operates in the latent space, we utilize **another latent encoder**, denoted as $\mathrm{LE}_{\mathrm{DM}}$, with the same structure as LE. $\mathrm{LE}_{\mathrm{DM}}$ compresses the blurry image $\mathbf{I}_{Blur}$ into latent space to get the condition latent $\mathbf{c} \in \mathbb{R}^{N \times C'}$. The denoising network predicts the noise conditioned on the $\mathbf{z}_t$ and $\mathbf{c}$, *i.e.*, $\boldsymbol{\epsilon}_\theta(\mathbf{z}_t, \mathbf{c}, t)$. With the substitution of $\boldsymbol{\epsilon}_\theta$ in Eq. (5) and set the variance to $(1{-}\alpha_t)$, we get:

$$\boldsymbol{z}_{t-1} = \frac{1}{\sqrt{\alpha_t}}(\boldsymbol{y}_t - \frac{1-\alpha_t}{\sqrt{1-\bar{\alpha}_t}}\boldsymbol{\epsilon}_\theta(\mathbf{z}_t, \mathbf{c}, t)) + \sqrt{1-\alpha_t}\boldsymbol{\epsilon}_t, \tag{6}$$

where $\boldsymbol{\epsilon}_t \sim \mathcal{N}(0, \mathbf{I})$. By iteratively sampling $\mathbf{z}_t$ using Eq. (6) $T$ times, we can generate the predicted prior feature $\hat{\mathbf{z}} \in \mathbb{R}^{N \times C'}$, as shown in Fig. 1(b). The predicted prior feature is then used to guide Transformer, that is, $\mathbf{z}_1{=}\hat{\mathbf{z}}$ in Fig. 1(c). Notably, since the distribution of the latent space ($\mathbb{R}^{N \times C'}$) is much simpler than that of images ($\mathbb{R}^{H \times W \times C}$) [50, 51], the prior feature can be generated with a small number of iterations. We further explore the iteration numbers $T$ in Sec. 4.2.

**Training Strategy.** Training DM means training denoising network $\boldsymbol{\epsilon}_\theta$. Previous works [15, 41] train the model by optimizing the weighted variational bound. The training objective is:

$$\nabla_{\boldsymbol{\theta}} \|\boldsymbol{\epsilon} - \boldsymbol{\epsilon}_\theta(\sqrt{\bar{\alpha}_t}\mathbf{z} + \sqrt{1-\bar{\alpha}_t}\boldsymbol{\epsilon}, \mathbf{c}, t)\|_2^2, \tag{7}$$

where $\mathbf{z}$ and $\mathbf{c}$ are prior feature and condition latent defined above; $t \in [1, T]$ is a random time-step; $\boldsymbol{\epsilon} \sim \mathcal{N}(0, \mathbf{I})$ denotes sampled noise. However, the objective in Eq. (7) only trains DM. Since the slight deviation between the predicted prior feature and the actual prior $\mathbf{z}$, directly combining DM with Transformer could cause a mismatch, which restricts the deblurring performance.

To overcome this issue, we jointly train the diffusion model and Transformer, inspired by previous work [51]. In this approach, for each training iteration, we use the prior feature $\mathbf{z}$ to generate the noise sample $\mathbf{z}_T$ through Eq. (4). As the time-step $T$ is small in latent space, we then run the complete $T$ iteration reverse processes (Eq. (6)) to generate the predicted prior feature $\hat{\mathbf{z}}$. The $\hat{\mathbf{z}}$ is used to guide Transformer through HIM ($\mathbf{z}_1{=}\hat{\mathbf{z}}$). Finally, the training loss is given by the sum of the deblurring loss $\mathcal{L}_{\text{deblur}}$ and the diffusion loss $\mathcal{L}_{\text{diffusion}}$, where the diffusion loss $\mathcal{L}_{\text{diffusion}}{=}\|\hat{\mathbf{z}} - \mathbf{z}\|_1$.

### 3.3 Inference

After completing the two-stage training, given a blurry input image $\mathbf{I}_{Blur} \in \mathbb{R}^{H \times W \times 3}$, the HI-Diff first compresses $\mathbf{I}_{Blur}$ into a condition latent $\mathbf{c} \in \mathbb{R}^{N \times C'}$ via latent encoder $\mathrm{LE}_{\mathrm{DM}}$. Then the predicted prior feature $\hat{\mathbf{z}} \in \mathbb{R}^{N \times C'}$ is generated by the diffusion model. Specifically, the diffusion model iteratively executes the reverse process (defined in Eq. (6)) $T$ times starting from a randomly sampled Gaussian Noise ($\boldsymbol{\epsilon} \sim \mathcal{N}(0, \mathbf{I})$). Finally, the HI-Diff reconstructs the deblurred image $\mathbf{I}_{DB} \in \mathbb{R}^{H \times W \times 3}$ from the input image $\mathbf{I}_{Blur}$, using Transformer, which is enhanced by the prior feature $\hat{\mathbf{z}}$.

| Method | Prior | Multi-Scale | Joint-training | Params (M) | FLOPs (G) | PSNR (dB) | SSIM |
|---|---|---|---|---|---|---|---|
| Basline | ✗ | ✗ | ✗ | 19.13 | 117.25 | 31.96 | 0.9528 |
| Single-Guide | ✓ | ✗ | ✓ | 21.98 | 125.39 | 32.00 | 0.9534 |
| Split-Training | ✓ | ✓ | ✗ | 23.99 | 125.47 | 30.73 | 0.9434 |
| HI-Diff (ours) | ✓ | ✓ | ✓ | 23.99 | 125.47 | 32.24 | 0.9558 |

Table 1: Ablation study. We train and test models on the GoPro [28] dataset. Image size is $3 \times 256 \times 256$ to calculate FLOPs. Prior: the prior feature generated by the diffusion model; Multi-scale: the multi-scale prior feature for hierarchical integration (as opposed to single-scale); Joint-training: the diffusion model and Transformer are trained jointly in stage two.

## 4 Experiments

### 4.1 Experimental Settings

**Data and Evaluation.** Following previous image deblurring methods, we evaluate our method on synthetic datasets (GoPro [28] and HIDE [39]) and the real-world dataset (RealBlur [34] and RWBI [56]). The GoPro dataset contains 2,103 pairs of blurry and sharp images for training and 1,111 image pairs for testing. The HIDE dataset provides testing 2,025 images. The RealBlur dataset has two sub-set: RealBlur-J and RealBlur-R. Each sub-set consists of 3,758 training pairs and 980 testing pairs. The RWBI dataset contains 3,112 blurry images captured with different hand-held devices. For synthetic datasets, we train HI-Diff on the GoPro training set and test it on GoPro and HIDE. Moreover, we further test the GoPro-trained model on RealBlur and RWBI to evaluate the generalization of our method. For real-world datasets, we train and test HI-Diff on RealBlur datasets, following previous works [54, 44]. We adopt two common metrics: PSNR and SSIM [49].

**Implementation Details.** Our HI-Diff consists of two parts: Transformer and the latent diffusion model. For Transformer, without loss of generality, we apply Restormer [53], a 4-level encoder-decoder Transformer architecture. From level-1 to level-4, we set the number of Transformer blocks as [3,5,5,6], the number of channels as [48,96,192,384], and the attention heads as [1,2,4,8]. Besides, there are 4 blocks in the refinement stage. The channel expansion factor is 2.66. For the latent diffusion model, the token number $N$ is 16, and the channel dimension $C'$ is 256. The latent encoder contains $L=6$ residual blocks. We set the total time-step $T$ as 8, and the variance hyperparameters $\beta_{1:T}$ constants increasing linearly from $\beta_1=0.1$ to $\beta_T=0.99$.

**Training Settings.** We train our HI-Diff with Adam optimizer [19] with $\beta_1=0.9$ and $\beta_2=0.99$. For stage one, the total training iterations are 300K. The initial learning rate is set as $2 \times 10^{-4}$ and gradually reduced to $1 \times 10^{-6}$ with the cosine annealing [27]. Following previous work [53], we adopt progressive learning. Specifically, we set the initial patch size as 128 and the patch size as 64. We progressively update the patch size and batch size pairs to $[(160^2,40),(192^2,32),(256^2,16),(320^2,16),(384^2,8)]$ at iterations [20K,40K,60K,80K,100K]. For stage two, we adopt the same training settings as in stage one. Moreover, following previous works [53, 54], we apply random rotation and flips for data augmentation. We use PyTorch [31] to implement our models with 4 A100 GPUs.

### 4.2 Ablation Study

In this section, we study the effects of different designs of our proposed method. We conduct all experiments on the dataset GoPro [28]. The iterations for stages one and two are 100K, respectively. The image patch size and batch size are set as $192 \times 192$ and 32. When we calculate the FLOPs, the image size is $3 \times 256 \times 256$. The results are reported in Tab. 1 and Figs. 2 and 3.

**Effects of Diffusion Prior.** We construct a baseline model without prior generated from diffusion in the first row of Tab. 1, denoted as Baseline, which is actually the vanilla Restormer [53] architecture. For fair comparisons, Baseline adopts the same implementation and training settings as HI-Diff (ours, listed in the fourth row). Comparing Baseline and HI-Diff, we can discover that using the prior feature generated by the latent diffusion model (DM) yields a 0.28 dB improvement. It demonstrates the effectiveness of our proposed method. In addition, the HI-Diff, integrated with diffusion, only adds 4.86M Params and 8.22G FLOPs over Baseline. It reveals that performing the diffusion model in highly compact latent space is very efficient. Furthermore, we show the visual comparisons of the Baseline (without the prior feature) and HI-Diff in Fig. 2 (first row). Our HI-Diff, generates sharper textures and complete structures, compared with Baseline.

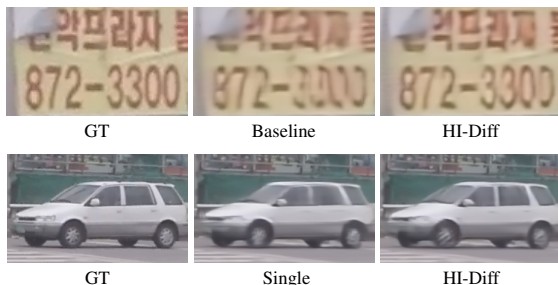

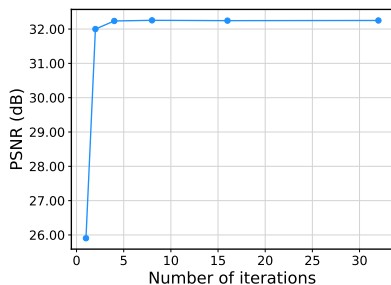

Figure 2: Deblurred samples for different models in Tab. 1. The first row shows effects of diffusion prior, while the second row exhibits effects of hierarchical integration.

Figure 3: Ablation study of the number of iterations $T$ in diffusion model, $T$: $\{1, 2, 4, 8, 16, 32\}$.

**Effects of Hierarchical Integration.** We conduct an ablation experiment on the hierarchical integration with the multi-scale prior feature (illustrated in Fig. 1(c)). We apply the single-scale prior feature on Transformer, which means we set $z_1=z_2=z_3=z$ (or $\hat{z}$). We denote the new model as Single-Guide, and its result is shown in the second row. We find that the PSNR of Single-Guide drops by 0.24dB compared with HI-Diff, which adopts the multi-scale prior feature. It indicates that the single-scale prior feature cannot adapt well to complex blurry situations. The visual comparison in Fig. 2 (second row) reveals that applying the hierarchical integration restores better-deblurred images.

**Effects of Joint Training Strategy.** We explore the impact of the joint training strategy in stage two. We train a model only optimized diffusion model in stage two, denoted as Split-Training. Specifically, for Split-Training, we first generate the prior feature $z$ from the ground truth and then apply the training objective defined in Eq. (7) to train the diffusion model alone. Then the diffusion model is directly combined with Transformer for evaluation after training is complete. For fair comparisons, Split-Training applies the same pre-trained (stage one) model as HI-Diff in stage two training, and the iterations are 100K. Comparing Split-Training and HI-Diff, HI-Diff is significantly better than Split-Training by 1.51 dB on PSNR value. These results are consistent with the analysis in Sec. 3.2 and demonstrate the importance of the joint training strategy.

**Effects of Iterations Number.** We further conduct an ablation study to investigate the influence of iteration numbers in the diffusion model. We set six different iterations $T$: $\{1, 2, 4, 8, 16, 32\}$ for the diffusion model. Meanwhile, the variance hyperparameters $\beta_{1:T}$ are always linearly interpolated from $\beta_1=0.1$ to $\beta_T=0.99$ for different $T$. We plot the PSNR of different iterations $T$ in Fig. 3. We find that only one iteration inference cannot generate the reasonable prior feature, limiting the deblurring performance. Besides, when the number of iterations reaches 8, the curve basically converges. It indicates that only a small number of iterations is needed to generate the suitable prior feature, since the simple distribution of the highly compact latent space (only contains $N=16$ tokens).

### 4.3 Evaluation on Synthetic Datasets

We compare our HI-Diff with 16 state-of-the-art methods: DeblurGAN [22], DeepDeblur [28], DeblurGAN-v2 [23], SRN [43], DBGAN [56], MT-RNN [30], DMPHN [55], SAPHN [42], SPAIR [32], MIMO-UNet+ [5], TTFA [4], MPRNet [54], HINet [2], Restormer [53], and Stripformer [44]. We show quantitative results in Tab. 2 and visual results in Fig. 4.

**Quantitative Results.** We train our method on GoPro [28], and test it on GoPro and HIDE [39]. Moreover, we further test the GoPro-trained model on the real-world dataset: RealBlur [34] (RealBlur-R and RealBlur-J). The PSNR/SSIM results on four benchmark datasets are reported in Tab. 2. Our method outperforms all compared state-of-the-art methods on all datasets.

When compared on synthetic datasets: GoPro and HIDE, our HI-Diff obtains the 0.25 dB gain on GoPro over Stripformer [44], the second best method. Meanwhile, compared with Restormer [53], the backbone of our method, our HI-Diff yields **0.41 dB** and **0.24 dB** gains on GoPro and HIDE.

When compared on real-world datasets: RealBlur-R and RealBlur-J, our HI-Diff exhibit a better generalization ability than other state-of-the-art algorithms. Compared with the recent best-performing method on GoPro, Stripformer, our method yields 0.33 dB on the RealBlur-J dataset. Besides, the HI-Diff outperforms the backbone, Restormer [53], by 0.09 dB on RealBlur-R and 0.19 dB on RealBlur-J. All these comparisons demonstrate the effectiveness of our HI-Diff.

| Method | GoPro [28] | | HIDE [39] | | RealBlur-R [34] | | RealBlur-J [34] | |
|---|---|---|---|---|---|---|---|---|
| | PSNR ↑ | SSIM ↑ | PSNR ↑ | SSIM ↑ | PSNR ↑ | SSIM ↑ | PSNR ↑ | SSIM ↑ |
| DeblurGAN [22] | 28.70 | 0.858 | 24.51 | 0.871 | 33.79 | 0.903 | 27.97 | 0.834 |
| DeepDeblur [28] | 29.08 | 0.914 | 25.73 | 0.874 | 32.51 | 0.841 | 27.87 | 0.827 |
| DeblurGAN-v2 [23] | 29.55 | 0.934 | 26.61 | 0.875 | 35.26 | 0.944 | 28.70 | 0.866 |
| SRN [43] | 30.26 | 0.934 | 28.36 | 0.915 | 35.66 | 0.947 | 28.56 | 0.867 |
| DBGAN [56] | 31.10 | 0.942 | 28.94 | 0.915 | 33.78 | 0.909 | 24.93 | 0.745 |
| MT-RNN [30] | 31.15 | 0.945 | 29.15 | 0.918 | 35.79 | 0.951 | 28.44 | 0.862 |
| DMPHN [55] | 31.20 | 0.940 | 29.09 | 0.924 | 35.70 | 0.948 | 28.42 | 0.860 |
| SAPHN [42] | 31.85 | 0.948 | 29.98 | 0.930 | N/A | N/A | N/A | N/A |
| SPAIR [32] | 32.06 | 0.953 | 30.29 | 0.931 | N/A | N/A | 28.81 | 0.875 |
| MIMO-UNet+ [5] | 32.45 | 0.957 | 29.99 | 0.930 | 35.54 | 0.947 | 27.63 | 0.837 |
| TTFA [4] | 32.50 | 0.958 | 30.55 | 0.935 | N/A | N/A | N/A | N/A |
| MPRNet [54] | 32.66 | 0.959 | 30.96 | 0.939 | 35.99 | 0.952 | 28.70 | 0.873 |
| HINet [2] | 32.71 | 0.959 | 30.32 | 0.932 | 35.75 | 0.949 | 28.17 | 0.849 |
| Restormer [53] | 32.92 | 0.961 | 31.22 | 0.942 | 36.19 | 0.957 | 28.96 | 0.879 |
| Stripformer [44] | 33.08 | 0.962 | 31.03 | 0.940 | 36.08 | 0.954 | 28.82 | 0.876 |
| HI-Diff (ours) | 33.33 | 0.964 | 31.46 | 0.945 | 36.28 | 0.958 | 29.15 | 0.890 |

Table 2: Quantitative comparisons on the four benchmark datsets: GoPro [28], HIDE [39], and RealBlur [34] (RealBlur-R and RealBlur-J). All models are trained only on GoPro dataset. Best and second best results are colored with red and blue. Our HI-Diff outperforms state-of-the-art methods.

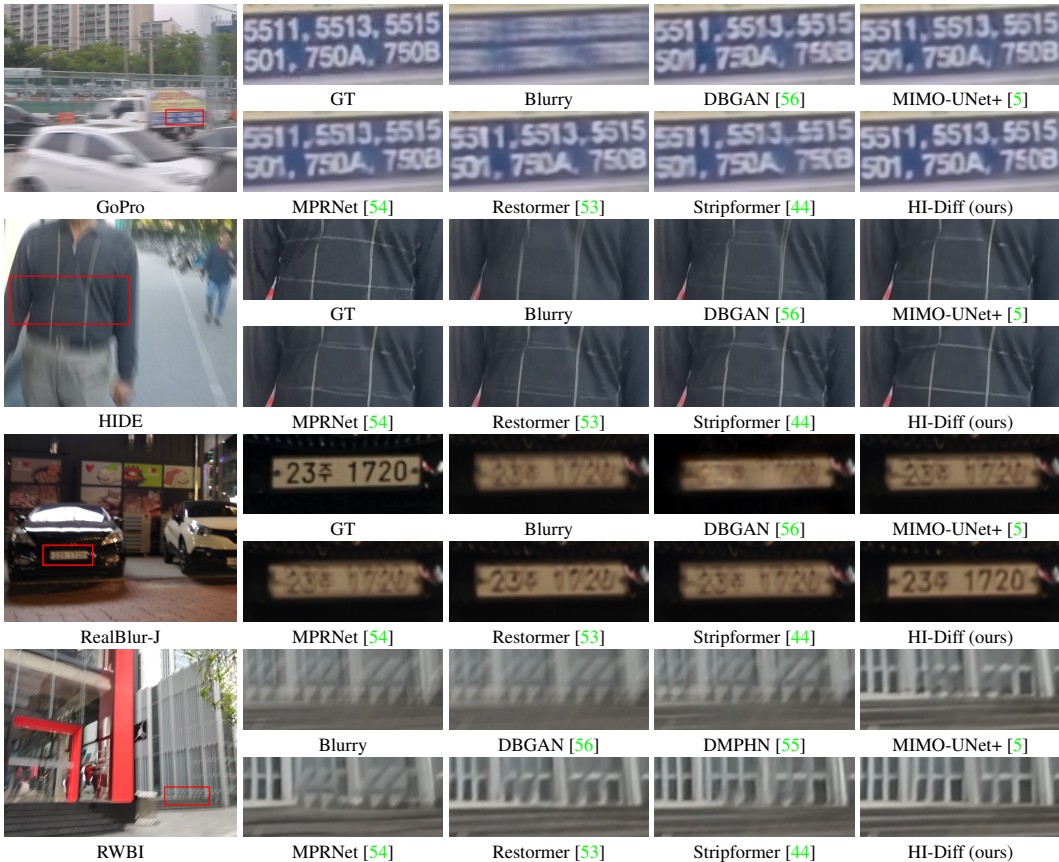

Figure 4: Visual comparison on GoPro [28], HIDE [39], RealBlur [34], and RWBI [56] datasets. RWBI only contains blurry images are captured with different hand-held devices. Models are trained only on the GoPro dataset. Our HI-Diff generates images with clearer details.

**Visual Results.** We show visual comparisons on GoPro [28], HIDE [39], RealBlur [34], and RWBI [56] in Fig. 4. We can observe that most compared methods suffer from artifacts or still contain significant blur effects. In contrast, our method can reconstruct more accurate textures and sharper edges. For example, in the HIDE sample, compared methods fail to reconstruct the white lines in the cloth, while our method recover sharp textures. All these visual comparisons are consistent with quantitative results and further demonstrate the superiority of our method.

| Dataset | Method | DeblurGAN-v2 [23] | SRN [43] | MIMO-UNet+ [5] | MPRNet [54] | BANet [45] | Stripformer [44] | HI-Diff (ours) |
|---|---|---|---|---|---|---|---|---|
| RealBlur-R [34] | PSNR ↑ | 36.44 | 38.65 | N/A | 39.31 | 39.55 | 39.84 | 41.01 |
| | SSIM ↑ | 0.935 | 0.965 | N/A | 0.972 | 0.971 | 0.974 | 0.978 |
| RealBlur-J [34] | PSNR ↑ | 29.69 | 31.38 | 31.92 | 31.76 | 32.00 | 32.48 | 33.70 |
| | SSIM ↑ | 0.870 | 0.909 | 0.919 | 0.922 | 0.9230 | 0.929 | 0.941 |

Table 3: Quantitative comparisons on RealBlur [34]. All models are trained and tested on the corresponding datasets. Best and second best results are colored with red and blue.

| Method | DMPHN [55] | MIMO-UNet+ [5] | MPRNet [54] | HINet [2] | Restormer [53] | Stripformer [44] | HI-Diff (ours) | HI-Diff-2 (ours) |
|---|---|---|---|---|---|---|---|---|
| Params (M) | 21.70 | 16.11 | 20.13 | 88.67 | 26.13 | 19.71 | 28.49 | 23.99 |
| FLOPs (G) | 195.44 | 150.68 | 760.02 | 67.51 | 154.88 | 155.03 | 142.62 | 125.47 |
| PSNR (dB) | 31.20 | 32.45 | 32.66 | 32.71 | 32.92 | 33.08 | 33.33 | 33.28 |

Table 4: Model complexity comparisons. Params, FLOPs, and PSNR on GoPro are reported. When we calculate the FLOPs, the image size is set as $3 \times 256 \times 256$.

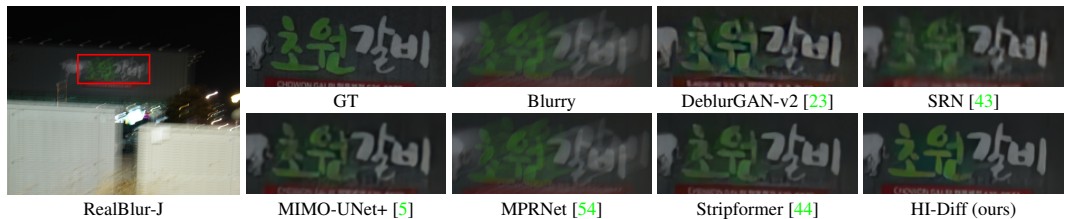

Figure 5: Visual comparison on the RealBlur [34] dataset. Models are trained on the RealBlur dataset.

## 4.4 Evaluation on Real-World Datasets

We further compare our HI-Diff with 6 state-of-the-art methods: DeblurGAN-v2 [23], SRN [43], MIMO-UNet+ [5], MPRNet [54], BANet [45], and Stripformer [44]. We show quantitative and visual results in Tab. 4 and Fig. 5. For fair comparisons, all previous method results are directly cited from the original papers or generated from official pre-trained models.

**Quantitative Results.** Table 3 reports PSNR/SSIM comparisons on real-world datasets: Real-Blur [34] (RealBlur-R and RealBlur-J). We train and test our HI-Diff on the RealBlur datasets, following previous works [54, 44]. Our method significantly outperforms other compared methods on the two datasets. Especially, compared with the recent best method, Stripformer, the HI-Diff obtains **1.17 dB** and **1.22 dB** gains on RealBlur-R and RealBlur-J, respectively.

**Visual Results.** We show visual comparisons on RealBlur in Fig. 5. Our method recovers sharper images with more high-frequency textures. However, most compared methods fail to recover clear images. For instance, compared methods have severe artifacts and blurring on green words, while our HI-Diff restores correct textures that are generally faithful to the ground truth. These visual results further demonstrate the strong ability of our HI-Diff for realistic image deblurring.

## 4.5 Model Size Analyses

We further show the comparison of model size (*e.g.*, Params) and computational complexity (*e.g.*, FLOPs) in Tab. 4. The FLOPs are measured when the image size is set as $3 \times 256 \times 256$. It shows that our HI-Diff has less FLOPs than CNN-based methods (*e.g.*, MPRNet [54]). Meanwhile, compared with Transformer-based methods, Restormer [53] and Stripformer [44], our HI-Diff performs better with comparable Params and less FLOPs. It indicates that our method achieves a better trade-off between performance and computational consumption. To further demonstrate the effectiveness of our method, we provide another variant of HI-Diff with less Params and FLOPs and better performance than Restormer. More details about HI-Diff-2 are provided in the supplementary material.

## 5 Conclusion

In this paper, we design the Hierarchical Integration Diffusion Model (HI-Diff), for realistic image deblurring. Specifically, HI-Diff performs the diffusion model to generate the prior feature for a regression-based method during deblurring. The regression-based method preserves the general

distribution, while the prior feature generated by the diffusion model enhances the details of the deblurred image. Meanwhile, the diffusion model is performed in a highly compact latent space with computational efficiency. Furthermore, we propose the hierarchical integration module (HIM) to fuse the prior feature and image features of Transformer hierarchically for better generalization ability in complex blurry scenarios. Extensive experiments on synthetic and real-world blur datasets demonstrate that our proposed HI-Diff outperforms state-of-the-art methods.

**Acknowledgments**. This work is partly supported by NSFC grant 62141220, 61972253, U1908212, 62271414, Science Fund for Distinguished Young Scholars of Zhejiang Province (LR23F010001), Research Center for Industries of the Future at Westlake University.

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
