# Supplementary Material: Hierarchical Integration Diffusion Model for Realistic Image Deblurring

**Zheng Chen**[1], **Yulun Zhang**[2*], **Ding Liu**[3], **Bin Xia**[4], **Jinjin Gu**[5,6], **Linghe Kong**[1*], **Xin Yuan**[7]

[1]Shanghai Jiao Tong University, [2]ETH Zürich, [3]Bytedance Inc, [4]Tsinghua University,
[5]Shanghai AI Laboratory, [6]The University of Sydney, [7]Westlake University

## 1 Model Variants

We provide more versions of HI-Diff to demonstrate the effectiveness of our proposed method. **First**, we provide a variant of HI-Diff, called HI-Diff-2, with less Params and FLOPs than Restormer [19]. **Second**, we fined-tune our HI-Diff with perceptual loss and adversarial loss, and obtain the HI-Diff-PE, with better performance in terms of perceptual quality.

| Method | Params (M) | FLOPs (G) | GoPro [10] | | HIDE [12] | |
|---|---|---|---|---|---|---|
| | | | PSNR ↑ | SSIM ↑ | PSNR ↑ | SSIM ↑ |
| Restormer [19] | 26.13 | 154.88 | 32.92 | 0.961 | 31.22 | 0.942 |
| Stripformer [15] | 19.71 | 155.03 | 33.08 | 0.962 | 31.03 | 0.940 |
| HI-Diff-2 (ours) | 23.99 | 125.47 | 33.28 | 0.964 | 31.36 | 0.945 |

Table 1: Quantitative comparisons on GoPro [10] and HIDE [12] (PSNR/SSIM). Best and second best results are colored with red and blue. Image size is $3 \times 256 \times 256$ to calculate FLOPs.

### 1.1 HI-Diff-2

**Implementation details.** We provide a variant of HI-Diff, called HI-Diff-2. For the 4-level encoder-decoder Transformer, we set the number of Transformer blocks as [3,5,5,6], the number of channels as [48,96,192,384], and the attention heads as [1,2,4,8]. The refinement stage contains 4 blocks. These settings are consistent with HI-Diff. Meanwhile, we set the channel expansion factor as 2. For the latent diffusion model, the token number $N$ and the channel dimension $C'$ are 16 and 256, respectively. The number of residual blocks in the latent encoder is $L$=5. The total time-step $T$ is set as 4 with $\beta_{1:T}$ increasing linearly from $\beta_1$=0.1 to $\beta_T$=0.99. We train HI-Diff-2 on GoPro [10] and test it on synthetic datasets: GoPro and HIDE [12]. The training settings are the same as for HI-Diff.

**Quantitative Results.** We compare the HI-Diff-2 with Restormer [19] and Stripformer [15]. The results are shown in Table 1. FLOPs are measured when the image size is set to $3 \times 256 \times 256$. As we can see, our HI-Diff-2 still performs better than Restormer and Stripformer on two datasets. Specifically, the HI-Diff-2 achieves 0.36 dB and 0.14 dB gains on GoPro and HIDE, compared with Restormer. Meanwhile, the HI-Diff-2 obtains 0.2 dB improvement over Stripformer on GoPro. Moreover, compared with Restormer, our HI-Diff-2 costs less Params and FLOPs by 2.14 M and 29.41 G. While compared with Stripformer, the HI-Diff-2 also costs less FLOPs. All these results further indicate the effectiveness of our proposed method.

---

*Corresponding authors: Yulun Zhang, yulun100@gmail.com; Linghe Kong, linghe.kong@sjtu.edu.cn

37th Conference on Neural Information Processing Systems (NeurIPS 2023).

| Method | LPIPS ↓ | DISTS ↓ | NIQE ↓ | PSNR ↑ | SSIM ↑ |
|---|---|---|---|---|---|
| MIMO-UNet+ [1] | 0.091 | 0.072 | 4.03 | 32.45 | 0.957 |
| MPRNet [20] | 0.089 | 0.075 | 4.09 | 32.66 | 0.959 |
| Restormer [19] | 0.084 | 0.072 | 4.11 | 32.92 | 0.961 |
| Stripformer [15] | 0.077 | 0.068 | 4.00 | 33.08 | 0.962 |
| DvSR [17] | 0.059 | - | 3.39 | 31.66 | 0.948 |
| HI-Diff-PE-1 (ours) | 0.051 | 0.031 | 3.53 | 33.27 | 0.963 |
| HI-Diff-PE-2 (ours) | 0.044 | 0.029 | 3.30 | 32.84 | 0.959 |

Table 2: Quantitative comparisons on **GoPro** [10]. Except for the distortion-based metrics: PSNR and SSIM, we evaluate methods on perceptual metrics: LPIPS [22], DISTS [2], and NIQE [9]. Best and second best results are colored with red and blue. HI-Diff-PE-1 and HI-Diff-PE-2 are obtained by fine-tuning HI-Diff. Our method obtains a better trade-off between perception and distortion.

## 1.2 HI-Diff-PE

**Implementation details.** In order to further improve the performance of HI-Diff on perceptual quality, we use perceptual loss and adversarial loss to fine-tune HI-Diff. For the perceptual loss, following previous work [16, 7, 8], we apply the VGG loss [5]:

$$\mathcal{L}_{\text{VGG}} = \|\Phi_{i,j}(\mathbf{I}_{GT}) - \Phi_{i,j}(G(\mathbf{I}_{Blur}))\|_1, \tag{1}$$

where $\mathbf{I}_{Blur}$ is the input blurry image, and $\mathbf{I}_{GT}$ represents its corresponding ground truth image; $G$ represents a generator network, which is the HI-Diff; $\Phi_{i,j}$ is the feature map obtained by the $j$-th convolutional layer before the $i$-th maxpooling layer within the VGG-19 network [13]. For the adversarial loss, we adopt the vanilla GAN objective [3, 7]:

$$\min_G \max_D \mathbb{E}_{\mathbf{I}_{GT} \sim p_{\text{train}}(\mathbf{I}_{GT})}[\log D(\mathbf{I}_{GT})] + \mathbb{E}_{\mathbf{I}_{Blur} \sim p_G(\mathbf{I}_{Blur})}[\log(1 - D(G(\mathbf{I}_{Blur})))], \tag{2}$$

where $G$ represents a discriminator. We denote the discriminator loss and generator loss as $\mathcal{L}_D$ and $\mathcal{L}_G$, respectively. Moreover, we still apply the $\mathcal{L}_1$ loss during the fine-tuning, where $\mathcal{L}_1 = \|G(\mathbf{I}_{Blur}) - \mathbf{I}_{GT}\|_1$. Therefore, the total loss for the generator (HI-Diff) is:

$$\mathcal{L}_{\text{total}} = \mathcal{L}_1 + \lambda \mathcal{L}_{\text{VGG}} + \eta \mathcal{L}_G, \tag{3}$$

where $\lambda$ and $\eta$ are the coefficients to balance difference loss terms.

Considering the distortion and perceptual quality are at odds with each other, we apply different fine-tuning settings to train two models, HI-DIff-PE-1 and HI-DIff-PE-2. We train two models on the GoPro [10] datasets. We fine-tune two models on GoPro [10]. For HI-DIff-PE-1, we use Adam optimizer [6] with $\beta_1$=0.9 and $\beta_2$=0.99 with learning rate as $1 \times 10^{-4}$. The $\lambda$ and $\eta$ are set as $1 \times 10^{-2}$ and $5 \times 10^{-3}$. The total training iterations are 1K with batch size 8 and patch size 384×384. For HI-DIff-PE-2, we apply Adam optimizer with a learning rate as $1 \times 10^{-4}$. Both $\lambda$ and $\eta$ are set as 1. The iterations are 50K, and the batch size and patch size are 8 and 384×384.

**Quantitative Results.** We compare the HI-Diff-PE-1 and HI-Diff-PE-2 with 5 state-of-the-art methods: MIMO-UNet+ [1], MPRNet [20], Restormer [19], Stripformer [15], and DvSR [17]. We test models on GoPro [10]. Except for the distortion-based metrics: PSNR and SSIM, we further evaluate methods on three perceptual metrics: LPIPS [22], DISTS [2], and NIQE [9]. The results are listed in Tab. 2. Note that, DvSR is the method that adopts the diffusion model.

For HI-Diff-PE-1, our method outperforms all other methods, except for the DISTS value compared with DvSR. Compared with the second best method, Stripformer, our HI-Diff-PE-1 yields 0.19 dB gain on PSNR, while reducing LPIPS by 0.026. Meanwhile, our HI-Diff-PE-1 achieves 0.35 dB gains on PSNR over Restormer. Besides, compared with the diffusion model, DvSR, the HI-Diff-PE-1 obtains a 1.61 dB improvement in PSNR and a 0.008 reduction in LPIPS.

For HI-Diff-PE-2, the model achieves the best performance in terms of perceptual metrics. Compared with HI-Diff-PE-1, HI-Diff-PE-2 adopts perceptual loss and adversarial loss with higher weight for more iterations of fine-tuning. It makes HI-Diff-PE-2 pay more attention to perceptual quality, while PSNR and SSIM decrease slightly. Compared to Restormer, HI-Diff-PE-2 reduces LPIPS by 47%, while reducing PSNR by only 0.08 dB. Moreover, compared with the diffusion model, DvSR, HI-Diff-PE-2 yields 0.18 gains on PSNR with better perceptual scores.

All these results indicate that our method obtains a better trade-off regarding distortion and perception.

| Method | Params | FLOPs | LPIPS ↓ | DISTS ↓ | NIQE ↓ | PSNR ↑ | SSIM ↑ |
|---|---|---|---|---|---|---|---|
| DvSR (CVPR'22) | 26.07M | 170.31T | 0.059 | N/A | 3.39 | 31.66 | 0.948 |
| DvSR-SA (CVPR'22) | 26.07M | 3406.22G | 0.078 | N/A | 4.07 | 33.23 | 0.963 |
| DiffIR (ICCV'23) | 26.94M | 120.99G | 0.081 | 0.071 | 4.13 | 33.20 | 0.963 |
| HI-Diff (ours) | 28.49M | 142.62G | 0.080 | 0.071 | 4.12 | 33.33 | 0.964 |
| HI-Diff-PE-1 (ours) | 28.49M | 142.62G | 0.051 | 0.031 | 3.53 | 33.27 | 0.963 |
| HI-Diff-PE-2 (ours) | 28.49M | 142.62G | 0.044 | 0.029 | 3.30 | 32.84 | 0.959 |

Table 3: Quantitative comparisons with diffusion models on **GoPro** [10]. We evaluate methods on two distortion-based metrics: PSNR and SSIM, and three perceptual metrics: LPIPS [22], DISTS [2], and NIQE [9]. The FLOPs are calculated when the input image size is set to $3 \times 256 \times 256$. Best and second best results are colored with red and blue.

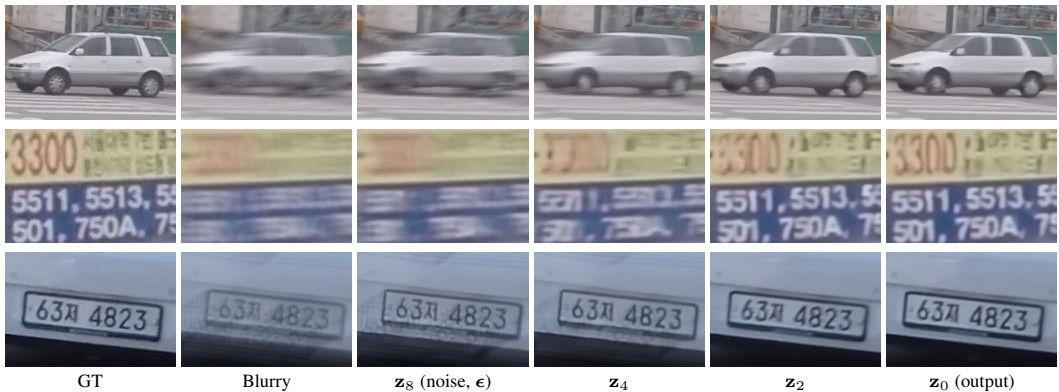

| GT | Blurry | $\mathbf{z}_8$ (noise, $\boldsymbol{\epsilon}$) | $\mathbf{z}_4$ | $\mathbf{z}_2$ | $\mathbf{z}_0$ (output) |

Figure 1: Visualization of the diffusion process. Notably, images in the rightmost column ($\mathbf{z}_0$) are actually the final outputs of our HI-Diff.

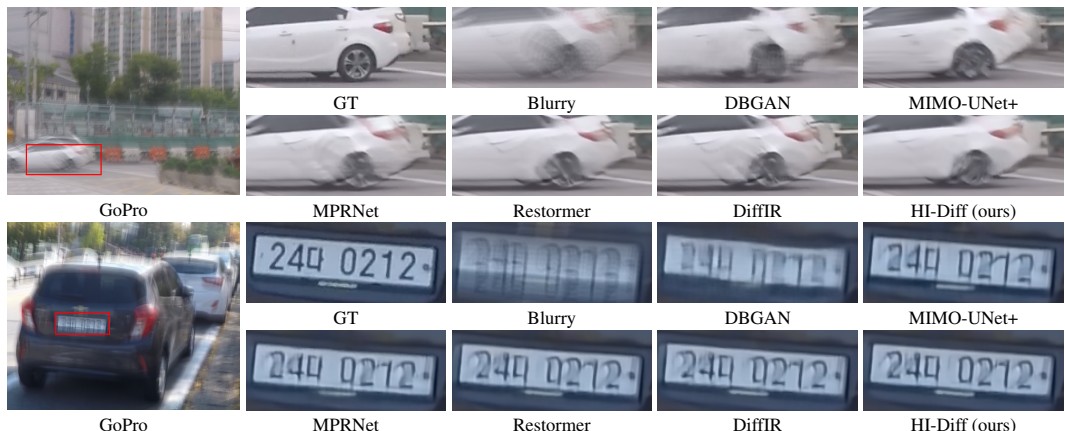

Figure 2: Failure case. Compared with other methods, Our HI-Diff can alleviate more artifacts.

## 2 More Quantitative Results

We compare our methods with some diffusion models: DvSR [17] and DiffIR [18] in Tab. 3. DvSR-SA represents DvSR applying sample average for distortion accuracy. Our methods achieve better performance on both distortion-based and perceptual metrics compared with other diffusion models.

## 3 More Visual Results

We show some visualization of the diffusion process in Fig. 1 and some failure cases in Fig. 2. Meanwhile, we provide more visual comparisons in Figs. 3, 4, 5, 6, 7, and 8.

### 3.1 Visualization of Diffusion Process

We provide some visualization in Fig. 1. The diffusion model generates the prior feature $z_0$ starting from a Gaussian Noise $z_8$ ($\epsilon \sim \mathcal{N}(0, I)$) through $T=8$ steps. We visualize some intermediate priors $z_t$ ($t \in [0, 8]$) to show the process, *i.e.*, $z_8$, $z_4$, $z_2$, and $z_0$. Since directly visualizing latent priors $z_t$ is not intuitive, we separately visualize the output (deblurred) images of Transformer guided by these priors. Notably, images in the rightmost column ($z_0$) are actually the final outputs of our HI-Diff.

We find that the blurred image gradually becomes sharp as the reverse process proceeds. Meanwhile, when the prior is noise (*i.e.*, $z_8$), the output of Transformer is not noise. This may be because Transformer features actively ignore the invalid priors through cross-attention when fused with priors.

### 3.2 Failure case

Our method HI-Diff also generates unpleasing artifacts (see the first case) or fails to recover sharp structures (see the second case) in some challenging cases. We provide some failure cases in Fig. 2. However, since we hierarchically integrate the diffusion model and Transformer, compared with other methods, our HI-Diff can alleviate more artifacts in the restoration results.

### 3.3 Synthetic Datasets

We show more visual comparisons on GoPro [10], HIDE [12], and RealBlur [11] in Figs. 3, 4, 5, 6 and 7. All models are trained only on GoPro. Most compared methods cannot recover accurate details and suffer from blurring artifacts in some challenging cases. In contrast, our HI-Diff handles several challenging cases better and recovers more realistic details. For instance, in the first GoPro sample in Fig. 3, our HI-Diff recovers clearer red shoes, while compared methods suffer blurring artifacts. In the second GoPro sample in Fig. 4, most compared methods fail to restore the white border. However, our HI-Diff recovers it clearly. Similar observations are shown in other images.

### 3.4 Real-World Datasets

We show more visual comparisons on RealBlur [11] in Fig. 8. All models are trained on RealBlur. Our HI-Diff handles several challenging cases better and recovers more details than compared methods. For instance, in the first GoPro sample in Fig. 8, our HI-Diff restores more textures (white horizontal lines) than other compared methods. These visual comparisons further demonstrate that our method has the powerful capability for realistic image deblurring.

## 4 Explanations for Checklist

### 4.1 Source Code

We provide code and trained models at `https://github.com/zhengchen1999/HI-Diff`.

### 4.2 Limitations

In this work, we implement our models with two-stage training. Compared to one-stage training, two-stage training is slightly tedious. Furthermore, we apply the vanilla diffusion model (DM) [4] and focus on the integration between the DM and the regression-based model. For other advanced DMs with better noise schedules and sampling strategies, we have not investigated yet.

### 4.3 Potential Negative Societal Impacts

We believe that our HI-Diff promotes the application of the DM in image deblurring. Our method benefits both academia and industry. We think there have a few potential negative societal impacts.

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

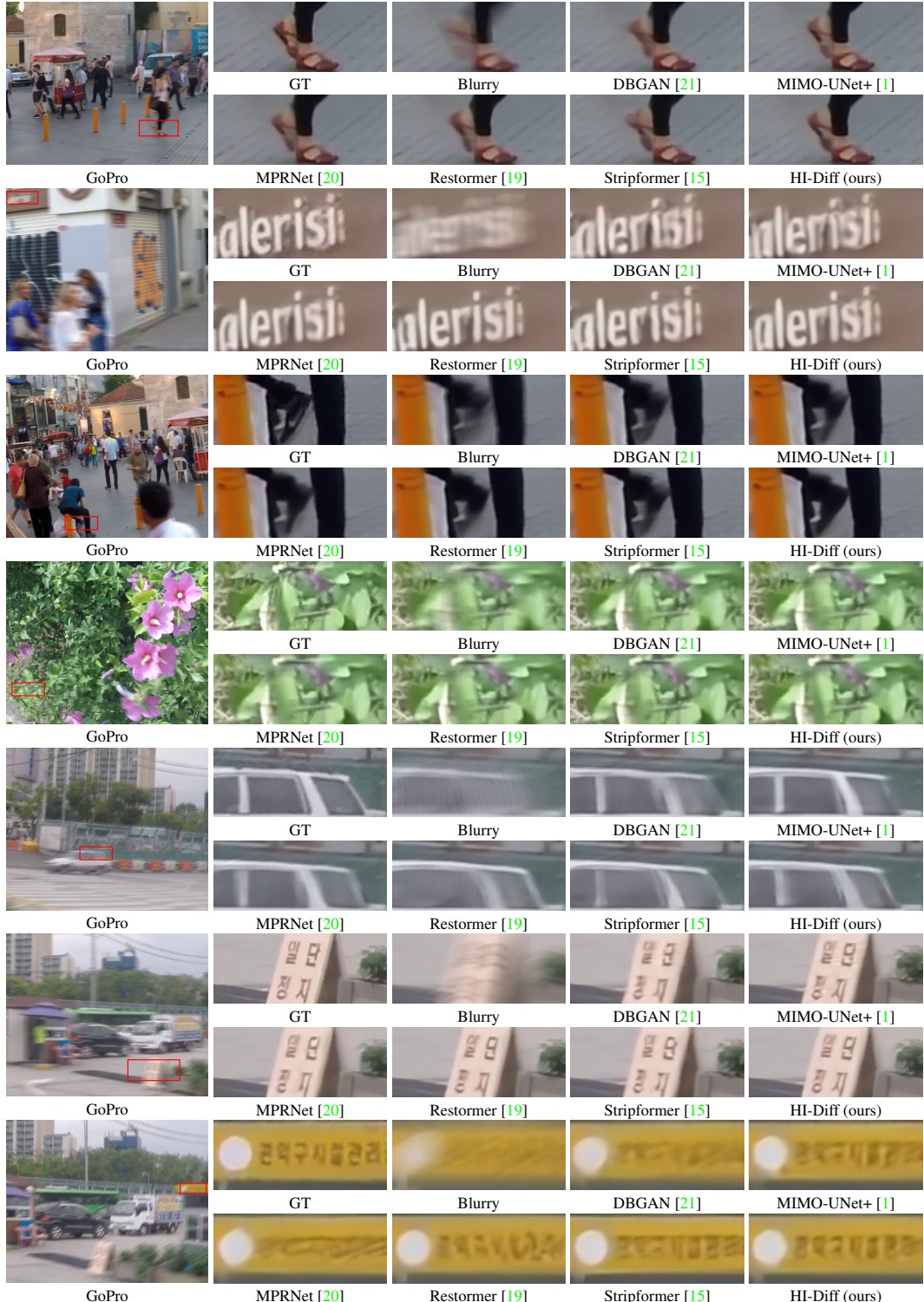

Figure 3: Visual comparison on the **GoPro** [10] dataset. Models are trained on the GoPro dataset.

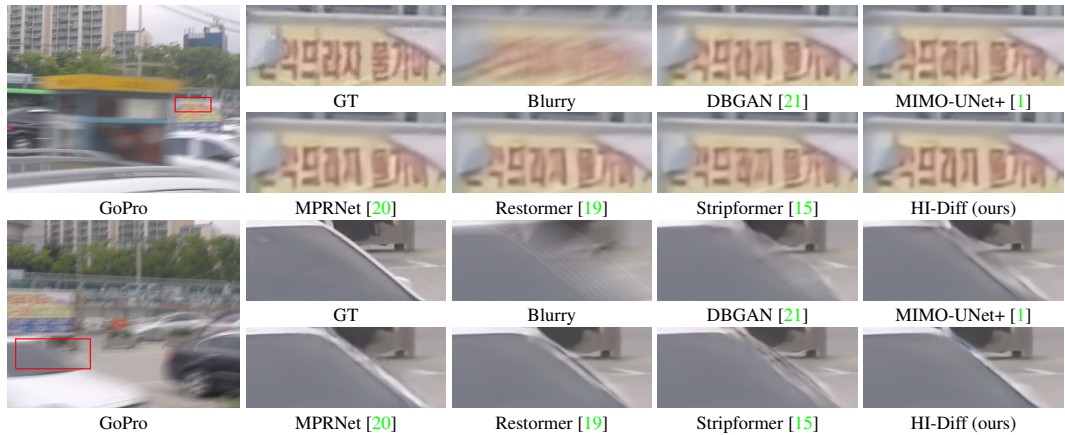

Figure 4: Visual comparison on the **GoPro** [10] dataset. Models are trained on the GoPro dataset.

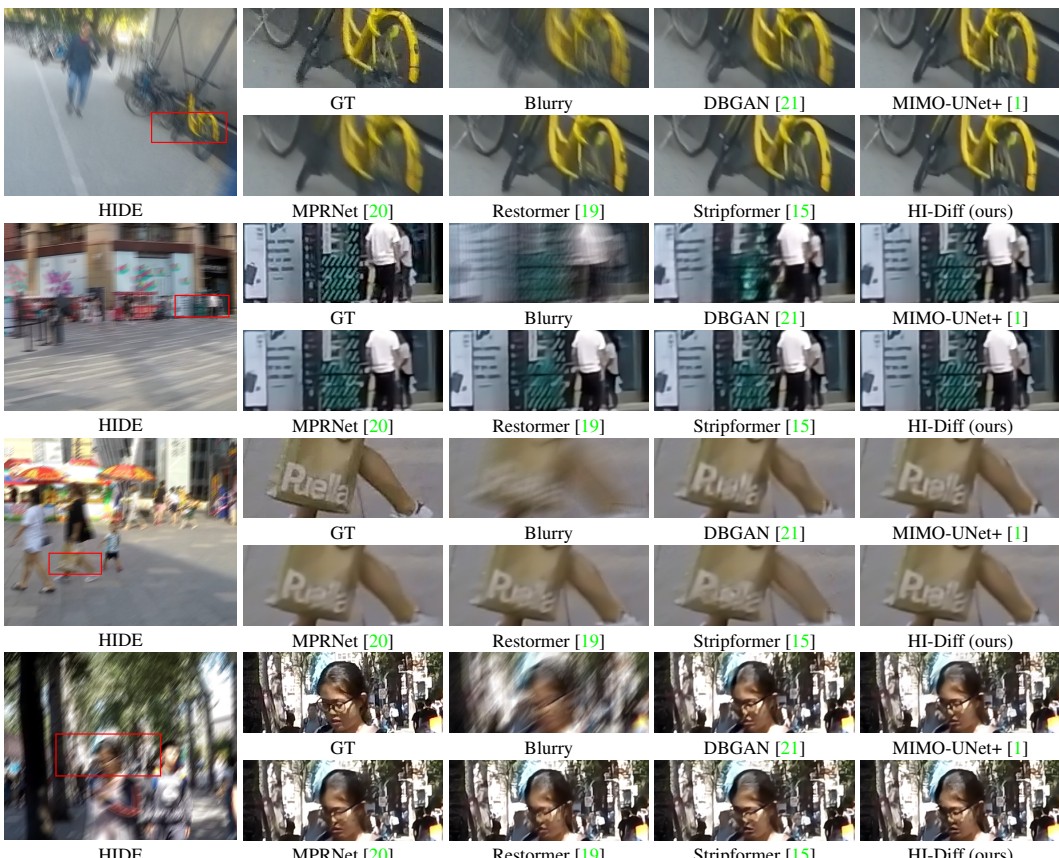

Figure 5: Visual comparison on the **HIDE** [12] dataset. Models are trained on the GoPro dataset.

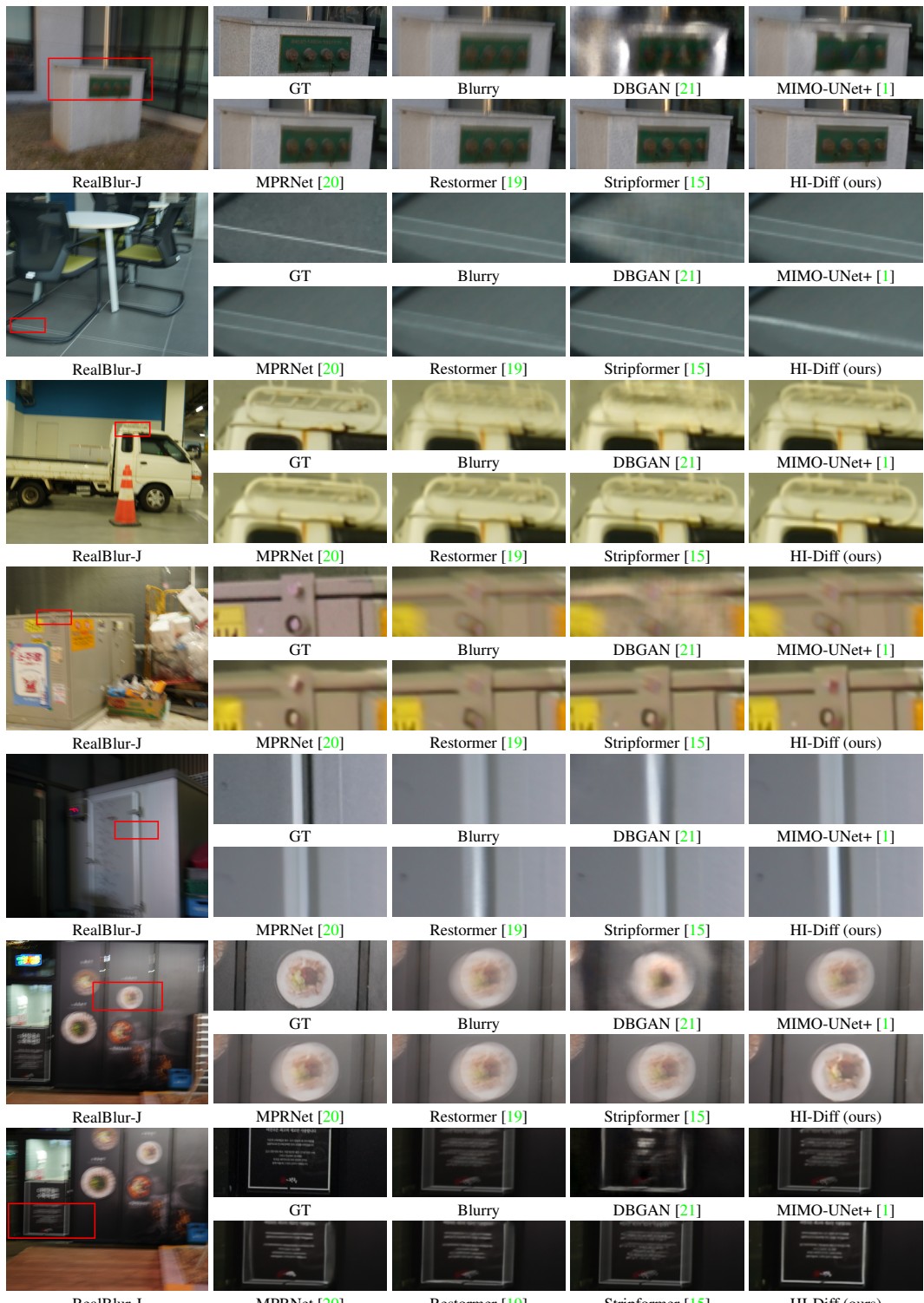

Figure 6: Visual comparison on the **RealBlur** [11] dataset. Models are trained on the GoPro dataset.

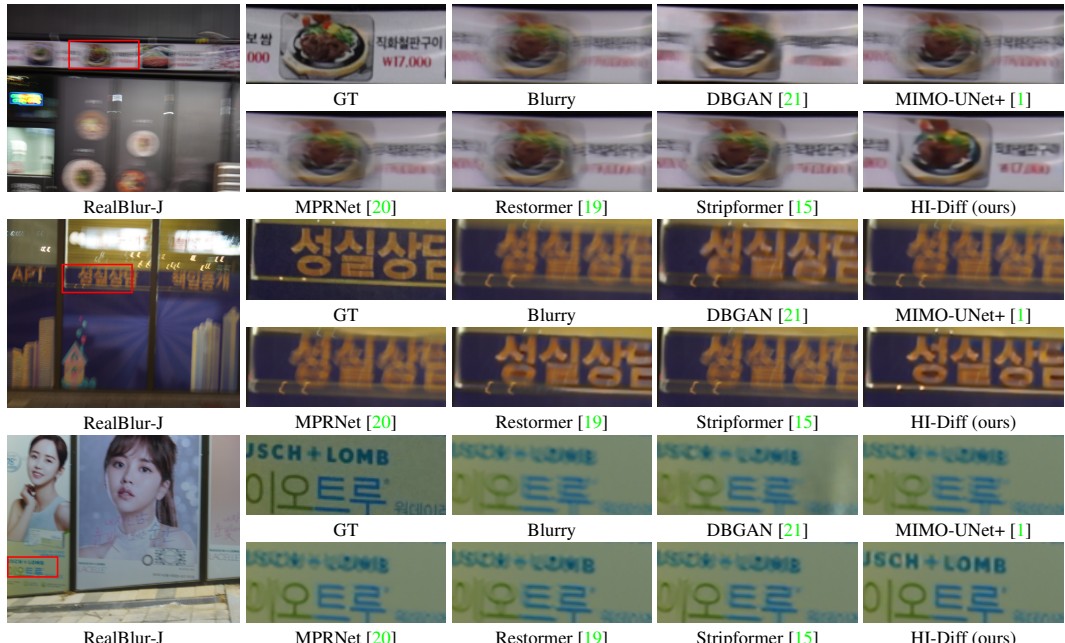

Figure 7: Visual comparison on the **RealBlur** [11] dataset. Models are trained on the GoPro dataset.

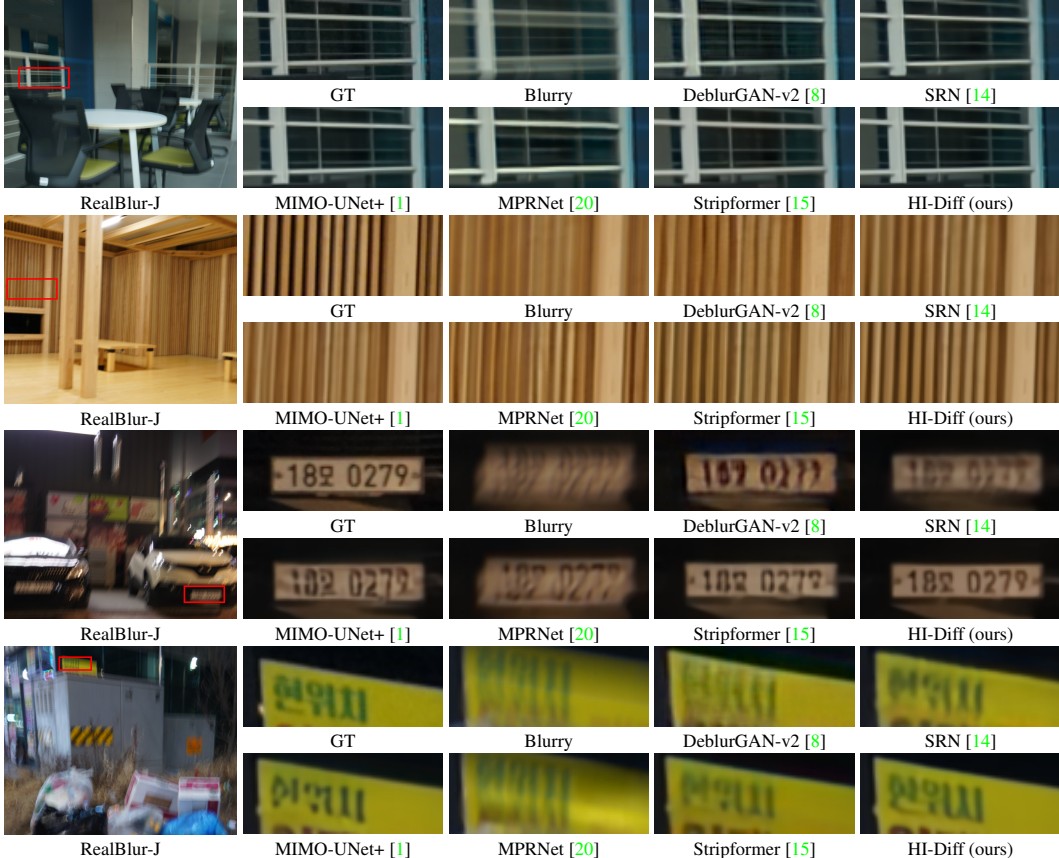

Figure 8: Visual comparison on the **RealBlur** [11] dataset. Models are trained on RealBlur.