# OpenReview forum: "Hierarchical Integration Diffusion Model for Realistic Image Deblurring"
_NeurIPS.cc/2023/Conference — NeurIPS 2023 spotlight_

### Official Review · Reviewer_w6Fg · 2023-07-01

**Soundness:** 3 good
**Presentation:** 3 good
**Contribution:** 2 fair
**Rating:** 6
**Confidence:** 3

**Summary:**

This paper proposed a novelty Hierarchical Integration Diffusion Model for deblurring task. By incorporation multi-scale latent priors, the proposed method achieved SOTA performance on both synthetic and real-world datasets.

**Strengths:**

1. The proposed method achieved SOTA performance.
2. The ablation study is very complete and the paper is easy to understand.

**Weaknesses:**

1. I think this paper is meaningful. But I argue that the authors should add more discussion about the difference between DiffIR. In my understanding, the difference is mainly using multi-scale latent prior, which has some novelty but seems small. Also in the introduction section, the authors state the motivation as "since the advantages of regression-based methods in distortion accuracy, we integrate DMs and regression-based methods", which seems similar with DiffIR.
2. Experiment comparison. This paper seems highly inspried by DiffIR. I suggest the authors also compare the results with the re-trained DiffIR model. The compare methods do not contain any diffusion models. In section 2.2, DvSR seems handle deblurring task using DM. I think the authors should add more comparisions.

**Questions:**

1. What are the difference between the proposed method and DiffIR? I think only using multi-scale latent space z seems not very novelty enough.
2. I suggest authors add more DM-based compare method, e.g., retrained DiffIR and DvSR.


**Limitations:**

As state in the introduction part, the diffusion model may gengerate unpleasing artifacts in the restoration results. Does this method also have some fail cases?

---

> ### Author Rebuttal · Authors · 2023-08-08
>
> ## Response to Reviewer w6Fg (denoted as R4)
>
> `Q4-1:` The authors should add more discussion about the difference between DiffIR. The difference is mainly using multi-scale latent prior. Also, the motivation: "since the advantages of regression-based methods in distortion accuracy, we integrate DMs and regression-based methods", seems similar with DiffIR.
>
> `A4-1:` Thanks for your valuable suggestions. We clarify the differences between DiffIR, and the novelty of our work below.
>
> *Note: All models are tested on GoPro, and the input size is 3×256×256 to calculate FLOPs.*
>
> **For "the difference is mainly using multi-scale latent prior".** The multi-scale latent prior is part of our novelty.
>
> 1. In general, the difference between HI-Diff and DiffIR is the integration approach between DM and Transformer. We design the **hierarchical integration**: **multi-scale prior** and **cross-attention interaction**, which is more suitable for non-uniform deblurring in real scenarios.
>
> 2. **The prior is different.** HI-Diff applies the multi-scale prior, while DiffIR uses the single-scale prior. The multi-scale prior adapts to encoder-decoder Transformer architecture for better integration. The ablation in Tab. 1 demonstrates this point.
>
>    | Method       | Params (M) | FLOPs (G) | PSNR (dB) |  SSIM  |
>    | ------------ | :--------: | :-------: | :-------: | :----: |
>    | Single-scale |   21.98    |  125.39   |   32.00   | 0.9534 |
>    | Multi-scale  |   23.99    |  125.47   |   32.24   | 0.9558 |
>
> 3. **The interaction is different.** HI-Diff adopts cross-attention in the hierarchical integration module (HIM), while DiffIR uses prior as dynamic modulation parameters. With cross-attention, the regions with varying degrees of blur in features can pay different attention to the prior, resulting in better (non-uniform) deblurring performance.
>
>    We replace cross-attention in HI-Diff with dynamic modulation and find that cross-attention yields better performance.
>
>    | Method             | Params (M) | FLOPs (G) | PSNR (dB) |  SSIM  |
>    | ------------------ | :--------: | :-------: | :-------: | :----: |
>    | dynamic modulation |   29.63    |  121.01   |   32.10   | 0.9544 |
>    | cross-attention    |   23.99    |  125.47   |   32.24   | 0.9558 |
>
>    Moreover, the **HIM** we proposed is **plug-and-play**, which is more convenient without modifying the components of the original model.
>
>
>
> **For "the motivation is similar with DiffIR".** The integration of DM and Transformer in our method is inspired by DiffIR. But our motivation is different from DiffIR.
>
> 1. **The motivation is different.** As described in the introduction, our motivation is threefold: reduce DM complexity, take advantage of Transformer, and better address the non-uniform blur. However, DiffIR focuses on improving the efﬁciency and stability of DM without **considering properties of realistic deblurring**.
>
> 2. **HI-Diff outperforms DiffIR.** We compare our HI-DIff with DiffIR. Our method performs better than DiffIR with comparable Params and FLOPs. It shows that our method, applying hierarchical integration, is more suitable for realistic non-uniform deblurring.
>
>    | Method           | Params (M) | FLOPs (G) | PSNR (dB) | SSIM  |
>    | ---------------- | :--------: | :-------: | :-------: | :---: |
>    | DiffIR (paper)   |   26.94    |  120.99   |   33.20   | 0.963 |
>    | HI-Diff (ours)   |   28.49    |  142.62   |   33.33   | 0.964 |
>    | HI-Diff-2 (ours) |   23.99    |  125.47   |   33.28   | 0.964 |
>
>
>
> `Q4-2:` Compare the results with the re-trained DiffIR model and DvSR.
>
> `A4-2:` Thanks for your suggestion. We provide more comparisons with diffusion models: **DvSR [48]** and **DiffIR [49]**.
>
> For DiffIR, apart from the results provided in the paper, we also **retrain the model** with the official code. Models are tested on GoPro, and the input size is 3×256×256 to calculate FLOPs.
>
> | Method             | Params |    FLOPs    | PSNR  | SSIM  |
> | ------------------ | :----: | :---------: | :---: | :---: |
> | DvSR               | 26.07M | 170.31**T** | 31.66 | 0.948 |
> | DiffIR (paper)     | 26.94M |   120.99G   | 33.20 | 0.963 |
> | DiffIR (retrained) | 26.94M |   120.99G   | 33.18 | 0.963 |
> | HI-Diff (ours)     | 28.49M |   142.62G   | 33.33 | 0.964 |
> | HI-Diff-2 (ours)   | 23.99M |   125.47G   | 33.28 | 0.964 |
>
> 1. Compared with the DvSR, our methods have much smaller FLOPs and better performance.
> 2. Compared with DiffIR, our methods perform better with comparable Params and FLOPs. This is because HI-Diff applies hierarchical integration, which is more suitable for processing non-uniform blur in real scenarios.
>
> **We provide more comprehensive comparisons in Tab. 1 of the PDF.**
>
>
>
> `Q4-3:` What are the difference between the proposed method and DiffIR? I think only using multi-scale latent space z seems not very novelty enough.
>
> `A4-3:` Thanks for asking for those details. We clarify them as follows.
>
> 1. The difference between our HI-Diff and DiffIR is the **hierarchical integration**: **multi-scale prior** and **cross-attention interaction**, which is more effective and suitable for non-uniform deblurring.
> 2. The multi-scale latent prior is part of our novelty.
>
> **We have responded to another similar question, `Q4-1`. Please refer to `A4-1` for more details.**
>
>
>
> `Q4-4:` I suggest authors add more DM-based compare method, e.g., retrained DiffIR and DvSR.
>
> `A4-4:` Thanks for the valuable suggestions. We compare more DM-based methods, e.g., DvSR [48] and DiffIR [49].
>
> Our HI-Diff outperforms other DM-based methods. **We have responded to another similar question, `Q4-2`. Please refer to `A4-2` for more details.**
>
>
>
> `Q4-5:` As state in the introduction part, the diffusion model may gengerate unpleasing artifacts in the restoration results. Does this method also have some fail cases?
>
> `A4-5:` Thanks for asking for those details. In our method, there are also some failure cases.
>
> We provide some cases in **Fig. 2 of the PDF**.

---

> > ### Comment · Reviewer_w6Fg · 2023-08-17
> >
> > Thank you for the time for rebuttal. My main considersions is this paper does not provide the comparison with other diffusion methods. The rebuttal has well sovled my considersions and thus I change my rate to weak accept after the rebuttal. After reading the review from Reviewer 5Ars, I suggest authors may also add some visual comparisons for deblur results with and without using the diffusion prior. Thank you.

---

> > > ### Author Response · Authors · 2023-08-17
> > > **Thanks Reviewer w6Fg for approving our work**
> > >
> > > Dear Reviewer w6Fg,
> > >
> > > Thanks for your response. We are happy to see that our response can solve your concerns.
> > >
> > > For "add some visual comparisons for deblur results with and without using the diffusion prior". In the ablation study (Sec 4.2), we have compared the model without the diffusion prior (**Baseline**), and the model with the diffusion prior (**HI-Diff, ours**). The quantitative results are provided in **Tab. 1** (first and fourth rows) of the main paper. And the visual comparisons are provided in **Fig. 2** (the first case). Meanwhile, thanks for your valuable suggestions. We will provide **more visual comparisons** in the revision.
> > >
> > > Best,
> > >
> > > Authors

---

### Official Review · Reviewer_5Ars · 2023-07-05

**Soundness:** 3 good
**Presentation:** 3 good
**Contribution:** 3 good
**Rating:** 7
**Confidence:** 5

**Summary:**

This paper presents the Hierarchical Integration Diffusion Model (HI-Diff) for realistic image deblurring. The HI-Diff utilizes diffusion models to generate multiscale priors in the latent space, which are integrated hierarchically into the deblurring process to improve the results.   Experiments are conducted on both synthetic and real-world blur datasets.

**Strengths:**

- A hierarchical integration module is proposed to fuse the prior into the model from multiple scales
- Experiments conducted on synthetic and real-world blur datasets demonstrate the state-of-the-art results, and the code would be released.

**Weaknesses:**

- It's unclear why authors adopt Diffusion Models (DMs) to model the prior. For one blurry image, the corresponding blurry prior should be deterministic instead of a distribution generated by DMs.
- The loss $L_{difussion}$ for the DMs is also weird (L202). If the DMs learn to produce the z, why not integrate z directly into the model?
- Tab. 1 and Fig.3 indicate that the DMs do not really matter. In Tab.1, compared to the baseline, it only improves by 0.04 dB. The gain of this method mainly comes from the multi-scale representation of the prior (> 0.2dB). Fig. 3, to some degree, also supports this. More iteration numbers (>5) in the diffusion model cannot improve the results.
- L222 L 243 seems to indicate the authors use the original Restormer [1]. But, with the additional DMs, the flops of this method are less than Restormer, as shown in Tab. 3. What is the difference compared to the original Restormer? Why not use the original setting?

[1] Restormer: Efficient transformer for high-resolution image restoration

**Questions:**

See the Weaknesses.

**Limitations:**

See the Weaknesses.

---

> ### Author Rebuttal · Authors · 2023-08-08
>
> ## Response to Reviewer 5Ars (denoted as R3)
>
> `Q3-1:` It's unclear why authors adopt Diffusion Models (DMs) to model the prior. For one blurry image, the corresponding blurry prior should be deterministic instead of a distribution generated by DMs.
>
> `A3-1:` Thanks for your question. We explain it below.
>
> 1. Compared with other methods, DMs have a stronger modeling ability to generate target priors. Therefore we apply DM to model the distribution of prior. Meanwhile, DM generates the prior **conditioned on** the blurry image, **not entirely randomly**.
>
> 2. Furthermore, the prior is not deterministic, since a blurry image may correspond to multiple sharp images (**ill-posed**). Therefore, some level of randomness in DM sampling is suitable.
>
> 3. We replace DM with Transformer in HI-Diff to generate priors. Models are tested on GoPro, and the input size is 3×256×256 to calculate FLOPs. Applying DM outperforms using Transformer.
>
>    | Method      | Params (M) | FLOPs (G) | PSNR (dB) |  SSIM  |
>    | ----------- | :--------: | :-------: | :-------: | :----: |
>    | Transformer |   24.84    |  125.48   |   32.09   | 0.9545 |
>    | DM          |   23.99    |  125.47   |   32.24   | 0.9558 |
>
>
>
> `Q3-2:` The loss $L_{diffusion}$ for the DMs is also weird (L202). If the DMs learn to produce the z, why not integrate z directly into the model?
>
> `A3-2:` Thanks for your question. We clarify them as follows.
>
> **For $L_{diffusion}$.**
>
> 1. The loss $L_{diffusion}$=$\Vert \hat{\mathbf{z}} - \mathbf{z} \Vert_1$, where $\mathbf{z}$ is prior from ground truth image and $\hat{\mathbf{z}}$ is the predicted prior after complete $T$ step reverse processes of DM. The loss is to allow DM to **learn to generate prior** directly.
> 2. For general DM (e.g., DDPM), it optimizes **one denoising step** at each training step. Therefore, it cannot generate the prior directly (needs total denoising steps). Thus, its training objective (Eq. (7) in our paper) is the noise ($\epsilon$). However, the final output of general DM is still the prior.
> 3. Different from general DM, we execute the **complete process** of DM at each training step, and let the DM directly learn to generate z. This is consistent with the final goal of general DM. Therefore, **the loss $L_{diffusion}$ is reasonable**.
>
> **For "why not integrate z directly into the model".**
>
> 1. It is feasible to integrate z directly into the (Transformer) model. However, this will cause Transformer to be performed at each denoising step. The overall **complexity is too high**, since Transformer is performed $T$ times.
>
> 2. Instead, we execute DM separately on the latent space, and only use the final result of DM for Transformer. Therefore, **the complexity is effectively reduced**, since Transformer is conducted once.
>
> 3. Furthermore, compared with methods (e.g., DvSR [48]) that execute the complete model at each step, our method (generate z separately) achieves better performance with much smaller FLOPs (evaluated on GoPro, the input size: 3×256×256 to calculate FLOPs).
>
>    | Method         | Params |    FLOPs    | PSNR  | SSIM  |
>    | -------------- | :----: | :---------: | :---: | :---: |
>    | DvSR (CVPR'22) | 26.07M | 170.31**T** | 31.66 | 0.948 |
>    | HI-Diff (ours) | 28.49M |   142.62G   | 33.33 | 0.964 |
>
>
>
> `Q3-3:` Tab. 1 and Fig.3 indicate that the DMs do not really matter. In Tab.1, compared to the baseline, it only improves by 0.04 dB. The gain of this method mainly comes from the multi-scale representation of the prior (> 0.2dB). Fig. 3, to some degree, also supports this. More iteration numbers (>5) in the diffusion model cannot improve the results.
>
> `A3-3:` Thanks for your question. We explain it as follows.
>
> **For "the DMs do not really matter".** We replace DM with Transformer in HI-Diff to generate priors, and the model performance decreases. It demonstrates that the DM is important.
>
> | Method      | Params (M) | FLOPs (G) | PSNR (dB) |  SSIM  |
> | ----------- | :--------: | :-------: | :-------: | :----: |
> | Transformer |   24.84    |  125.48   |   32.09   | 0.9545 |
> | DM          |   23.99    |  125.47   |   32.24   | 0.9558 |
>
> **For "the gain mainly comes from the multi-scale representation".**
>
> 1. Without multi-scale representation, the prior cannot be effectively fused in Transformer features, restricting performance. Thus, the gain is little.
> 2. The result of (Multi-Scale + Transformer) is lower than that of applying DM. This indicates that both DM and Multi-Scale are important. **Both two components should be considered together.**
>
> **For "more iteration numbers cannot improve the results".**
>
> 1. Since the prior is in the latent space, DM does not need too many iterations to model it. However, it doesn't mean DM isn't important.
> 2. Compared with 1000+ iterations in general DM (DDPM), fewer iterations (i.e., 8) further indicate the effectiveness of our proposed method.
>
>
>
> `Q3-4:` The authors use the original Restormer [1]. With the additional DMs, the flops of this method are less than Restormer (Tab. 3). What is the difference compared to the original Restormer? Why not use the original setting?
>
> `A3-4:` Thanks for your question. We explain it below.
>
> **For "the difference compared to Restormer".**
>
> 1. Our HI-Diff (model in Tab. 3 (4)) applies the original structure of Restormer with **fewer** block numbers in each stage. **Other settings remain the same.**
>
> 2. We provide a comparison between HI-Diff and Restormer. Models are tested on GoPro, and the input size is 3×256×256 to calculate FLOPs.
>
>    | Method    | Block in each stage | Params (M) | FLOPs (G) | PSNR (dB) |
>    | --------- | :-----------------: | :--------: | :-------: | :-------: |
>    | Restormer |    [4, 6, 6, 8]     |   26.13    |  154.88   |   32.92   |
>    | HI-Diff   |    [3, 5, 5, 6]     |   28.49    |  142.62   |   33.33   |
>
> **For "why not the original setting".** We apply fewer blocks to make the overall Params and FLOPs of HI-Diff close to Restormer to realize **a fair comparison**.

---

> > ### Author Response · Authors · 2023-08-17
> > **Follow-up discussions with Reviewer R3 (5Ars)**
> >
> > Dear Reviewer 5Ars,
> >
> > We thank you for your valuable review time and comments. We have responded to the related questions, which we believe have covered your concerns.
> >
> > 1. We explain the **reason** for applying the **diffusion model** (DM), and compare it with Transformer to demonstrate its **superiority**.
> > 2. We analyze the rationality of the **loss** for DM and the reasons for not **integrating z** directly into the Transformer model.
> > 3. We analyze and conduct experiments to demonstrate **the importance of DM**.
> > 4. We clarify the difference between our method and **Restormer**, and the reasons for not using the original setting.
> >
> > We hope to discuss further with you whether or not your concerns have been addressed. Please let us know if you still have any unsolved or other concerns. Then, we have enough time to provide further feedback. Thanks.
> >
> > Best,
> >
> > Authors

---

> > > ### Comment · Area_Chair_HRVD · 2023-08-18
> > >
> > > Reviewer 5Ars: Please read the rebuttal and comments by authors and respond ASAP.

---

> > > ### Comment · Reviewer_5Ars · 2023-08-19
> > >
> > > I appreciate your patient and comprehensive response. Your rebuttal effectively addresses the majority of my concerns. As a result, I will increase my rating to "Accept."

---

> > > > ### Author Response · Authors · 2023-08-19
> > > > **Thanks Reviewer 5Ars for approving our work**
> > > >
> > > > Dear Reviewer 5Ars,
> > > >
> > > > Thanks for your response. We are happy to see that our response can solve your concerns.
> > > >
> > > > The results and analyses corresponding to your questions further improve the quality of our work. Thank you!
> > > >
> > > > Best,
> > > >
> > > > Authors

---

### Official Review · Reviewer_ymRm · 2023-07-05

**Soundness:** 4 excellent
**Presentation:** 4 excellent
**Contribution:** 4 excellent
**Rating:** 8
**Confidence:** 5

**Summary:**

The authors propose a new image deblurring model, Hierarchical Integration Diffusion Model (HI-Diff). The HI-Diff uses the diffusion models to produce priors in a highly compacted latent space, and is integrated into the deblurring process hierarchically with the proposed hierarchical integration module (HIM). The effectiveness of each component is demonstrated by the ablation study. The main results on synthetic and real-world deblurring show that HI-Diff outperforms recent state-of-the-art methods.

**Strengths:**

1. The proposed HI-Diff is simple and effective. The idea that leveraging the power of the diffusion model and integrating it into the deblurring process hierarchically is reasonable and novel for image deblurring.

2. The design of the HI-Diff is stated clearly and logically. The authors expound the motivation and realization of the design in detail. And the effectiveness of each part is proved by the ablation study.

3. The main comparisons with recent state-of-the-art methods are extensive. The evaluation on both synthetic and real-world datasets demonstrates the superiority of the proposed methods.

4. In the supplementary material, the authors provide more variant models and more quantitative and qualitative comparisons, further revealing the promising performance of HI-DIff.

5. The paper is well-organized, and the writing is good and easy to read.

6. The authors also provide the code and pre-trained models for results reproduction. This reveals the solidness of the work and helps other researchers to follow.

**Weaknesses:**

1. The feature prior is the key of this work. However, the paper lacks a specific analysis of the prior. For example, the difference and impact of prior generated on different inputs.

2. As mentioned in the paper, the proposed method generates more realistic deblurred images. But the comparisons in Tabs. 2 and 3 are about distortion-based metrics (e.g., PSNR). More comparisons on perceptual metrics should be provided.

3. Although the authors provide the FLOPs and Params comparisons in Tab. 4, the latency (running time), another important indicator in low-level tasks, needs to be provided.

**Questions:**

1. Please clarify the differences between the HI-Diff and DiffIR [49].

2. Compare HI-Diff with more generative models (like GAN and Diffusion), and evaluate on perceptual metrics.

3. Provide the latency comparisons to further show the effectiveness of the method.

4. The authors conduct train models on synthetic and real-world datasets. Are the settings of two model the same? if different, clarify the details.

**Limitations:**

The limitations and potential negative societal impact of the work have been discussed in the supplementary material.

---

> ### Author Rebuttal · Authors · 2023-08-08
>
> ## Response to Reviewer ymRm (denoted as R2)
>
> `Q2-1:` The feature prior is the key of this work. However, the paper lacks a specific analysis of the prior.
>
> `A2-1:` Thanks for pointing it out. We provide an analysis of the prior.
>
> 1. We compare the similarity of priors generated on different inputs. We find that the similarity between priors (e.g., MSE) positively correlates with the similarity between input images (e.g., SSIM).
> 2. We further provide some visual results in **Fig. 1 of the PDF**, which intuitively show the impact of priors.
>
>
>
> `Q2-2:` More comparisons on perceptual metrics should be provided.
>
> `A2-2:` Thanks for your suggestion. We have provided a comparison of perceptual metrics in **Tab. 2** of the **supplementary material**. We also show part of them here (with more compared methods). Models are tested on GoPro.
>
> | Method              | Model | LPIPS $\downarrow$ | DISTS $\downarrow$ | NIQE $\downarrow$ | PSNR $\uparrow$ | SSIM $\uparrow$ |
> | ------------------- | :---: | :----------------: | :----------------: | :---------------: | :-------------: | :-------------: |
> | DBGAN (CVPR'20)     |  GAN  |       0.110        |       0.078        |       4.06        |      31.10      |      0.942      |
> | DvSR (CVPR'22)      |  DM   |       0.059        |        N/A         |       3.39        |      31.66      |      0.948      |
> | DiffIR (ICCV'23)    |  DM   |       0.081        |       0.071        |       4.13        |      33.20      |      0.963      |
> | HI-Diff (ours)      |  DM   |       0.080        |       0.071        |       4.12        |      33.33      |      0.964      |
> | HI-Diff-PE-1 (ours) |  DM   |       0.051        |       0.031        |       3.53        |      33.27      |      0.963      |
> | HI-Diff-PE-2 (ours) |  DM   |       0.044        |       0.029        |       3.30        |      32.84      |      0.959      |
>
> Our method achieves the best performance on both distortion-based and perceptual metrics.
>
> **We provide more comprehensive comparisons in Tab. 1 of the PDF.**
>
>
>
> `Q2-3:` Although the authors provide the FLOPs and Params comparisons in Tab. 4, the latency (running time), another important indicator in low-level tasks, needs to be provided.
>
> `A2-3:` Thanks for pointing it out. We provide the latency (i.e., running time) comparison. The running time is tested on one 3090 GPU with the input size of 3×256×256. We calculate the average time over 100 images. Our method achieves comparable running time with other methods.
>
> | Method       | MPRNet | Restormer | HI-Diff (ours) | HI-Diff-2 (ours) |
> | ------------ | :----: | :-------: | :------------: | :--------------: |
> | Latency (ms) | 77.46  |   82.05   |     75.89      |      65.98       |
>
>
>
> `Q2-4:` Please clarify the differences between the HI-Diff and DiffIR [49].
>
> `A2-4:` Thanks for asking for those details. We clarify them as follows.
>
> *Note: All models are tested on GoPro, and the input size is 3×256×256 to calculate FLOPs.*
>
> 1. In general, the difference between HI-Diff and DiffIR is the **hierarchical integration**: **multi-scale prior** and **cross-attention interaction**, which is more suitable for non-uniform deblurring.
>
> 2. **The prior is different.** HI-Diff applies the multi-scale prior, while DiffIR uses the single-scale prior. The multi-scale prior adapts to different scale features in encoder-decoder Transformer architecture for better integration.
>
>    | Method       | Params (M) | FLOPs (G) | PSNR (dB) |  SSIM  |
>    | ------------ | :--------: | :-------: | :-------: | :----: |
>    | Single-scale |   21.98    |  125.39   |   32.00   | 0.9534 |
>    | Multi-scale  |   23.99    |  125.47   |   32.24   | 0.9558 |
>
> 3. **The interaction is different.** HI-Diff adopts cross-attention, while DiffIR uses prior as dynamic modulation parameters. Features pay different attention to the prior with cross-attention, which is more suitable for non-uniform deblurring. Experiments demonstrate that applying cross-attention outperforms using dynamic modulation.
>
>    | Method             | Params (M) | FLOPs (G) | PSNR (dB) |  SSIM  |
>    | ------------------ | :--------: | :-------: | :-------: | :----: |
>    | dynamic modulation |   29.63    |  121.01   |   32.10   | 0.9544 |
>    | cross-attention    |   23.99    |  125.47   |   32.24   | 0.9558 |
>
> 4. **Experimental comparison.** Our method performs better than DiffIR with comparable Params and FLOPs. It shows that our method, applying hierarchical integration, is more suitable for realistic non-uniform deblurring.
>
>    | Method           | Params (M) | FLOPs (G) | PSNR (dB) | SSIM  |
>    | ---------------- | :--------: | :-------: | :-------: | :---: |
>    | DiffIR           |   26.94    |  120.99   |   33.20   | 0.963 |
>    | HI-Diff (ours)   |   28.49    |  142.62   |   33.33   | 0.964 |
>    | HI-Diff-2 (ours) |   23.99    |  125.47   |   33.28   | 0.964 |
>
>
>
> `Q2-5:` Compare HI-Diff with more generative models (like GAN and Diffusion), and evaluate on perceptual metrics.
>
> `A2-5:` Thanks for your suggestion. We compare HI-Diff with GAN: **DBGAN [54]**, Diffusion: **DvSR [48]** and **DiffIR [49]** on distortion-based and perceptual metrics. Our HI-Diff achieves the best results.
>
> **We have responded to another similar question, `Q2-2`. Please refer to `A2-2` for more details.**
>
>
>
> `Q2-6:` Provide the latency comparisons to further show the effectiveness of the method.
>
> `A2-6:` Thanks for your suggestion.
>
> We further compare the latency of HI-Diff with other methods. Our HI-Diff achieves a comparable running time with other methods. **We have responded to another similar question, `Q2-3`. Please refer to `A2-3` for more details.**
>
>
>
> `Q2-7:` The authors conduct train models on synthetic and real-world datasets. Are the settings of two model the same? if different, clarify the details.
>
> `A2-7:` Thanks for your question. The settings are the same for the two models on synthetic and real-world datasets.

---

> > ### Comment · Reviewer_ymRm · 2023-08-18
> > **My concerns are well addressed by the extensive results and analyses.**
> >
> > Thanks for providing such a detailed response. The authors provide extensive quantitative and visual results, which well solve my questions. The extensive explanations and analyses further figure out the unclear parts in my first-round review.
> > I also read other reviewers’ comments and the corresponding responses. Overall, I am very satisfied with the response. I would like to raise my score and vote for acceptance.

---

> > > ### Author Response · Authors · 2023-08-18
> > > **Thanks Reviewer ymRm for approving our work**
> > >
> > > Dear Reviewer ymRm,
> > >
> > > Thanks for your response. We are happy to see that our quantitative and visual results can solve your concerns.
> > >
> > > Best,
> > >
> > > Authors

---

### Official Review · Reviewer_mQcX · 2023-07-06

**Soundness:** 4 excellent
**Presentation:** 4 excellent
**Contribution:** 4 excellent
**Rating:** 7
**Confidence:** 4

**Summary:**

The paper introduces the Hierarchical Integration Diffusion Model (HI-Diff), a novel approach for realistic image deblurring. It combines a diffusion model and a regression-based model, performing the diffusion process in a compact latent space to generate informative priors for deblurring. These priors are integrated into the regression-based model using a hierarchical module that adapts to complex blurry scenarios. The paper also presents a two-stage training strategy to optimize the latent encoder and diffusion model together. Experimental results on synthetic and real-world datasets demonstrate the effectiveness, efficiency, and superiority of the proposed method over state-of-the-art techniques in terms of PSNR, SSIM, and visual quality.

**Strengths:**

- The paper presents a novel and effective approach for leveraging the diffusion model in image deblurring, yielding realistic details and mitigating unwanted artifacts.
- By executing the diffusion model within a compact latent space, the paper successfully reduces computational complexity, enabling faster inference with fewer iterations.
- Through comprehensive experiments on synthetic and real-world datasets, the paper demonstrates the superior performance of the proposed method compared to state-of-the-art approaches.

**Weaknesses:**

- The diffusion models have an advantage in generating high-quality image details. However, in this paper the diffusion model reconstruct feature in the compacted latent space. The paper does not show and explain the superiority of diffusion models over other models in this task.
- The paper does not conduct a comprehensive comparison with other diffusion models for image deblurring. These methods also apply diffusion models in different ways to address the challenges of image deblurring, such as computational efficiency, distortion accuracy, and generalization ability. It would be interesting to see how HI-Diff compares with these methods in terms of performance and complexity.
- The paper does not analyze the effectiveness of the latent compression and evaluate the hyperparameters. It would be helpful to provide some ablation studies or analysis on the latent compression encoder.
- Lack of one-stage ablation study or a clearer explanation is needed.

**Questions:**

-  Can you please elaborate on the structure and parameters of the diffusion model in more detail?
- It would be beneficial to include some visual results showcasing the effects of the diffusion process, if available.
- In the event of replacing the diffusion model with alternative restoration models like transformer-based approaches, would the proposed method still maintain its effectiveness?

**Limitations:**

Based on the available information on the web page, the paper does not explicitly address the limitations of their proposed method.

---

> ### Author Rebuttal · Authors · 2023-08-08
>
> ## Response to Reviewer mQcX (denoted as R1)
>
> `Q1-1:` The paper does not show and explain the superiority of diffusion models over other models in this task.
>
> `A1-1:` Thanks for pointing it out. We explain and conduct experiments to show the superiority of diffusion models (DMs).
>
> 1. **Explanation:** Compared with other methods, DMs have a stronger distribution (image / latent) modeling ability. Thus, DM can generate high-quality image details. Meanwhile, not limited to image distributions, it also applies to latent spaces. Therefore, we adopt DM to generate the prior in latent space.
>
> 2. **Experiments:** We replace DM with a Transformer-based model in HI-Diff to generate priors. We use the same training settings as HI-Diff for the new model. Models are tested on GoPro, and the input size is 3×256×256 to calculate FLOPs. Applying DM outperforms using Transformer.
>
>    | Method      | Params (M) | FLOPs (G) | PSNR (dB) |  SSIM  |
>    | ----------- | :--------: | :-------: | :-------: | :----: |
>    | Transformer |   24.84    |  125.48   |   32.09   | 0.9545 |
>    | DM          |   23.99    |  125.47   |   32.24   | 0.9558 |
>
>
>
>
> `Q1-2:` It would be interesting to see how HI-Diff compares with other diffusion models for image deblurring.
>
> `A1-2:` Thanks for your suggestions. We compare HI-Diff with other diffusion models: **DvSR [48]** and **DiffIR [49]**. Models are tested on GoPro, and the input size is 3×256×256 to calculate FLOPs.
>
> | Method           | Params |    FLOPs    | PSNR  | SSIM  |
> | ---------------- | :----: | :---------: | :---: | :---: |
> | DvSR (CVPR'22)   | 26.07M | 170.31**T** | 31.66 | 0.948 |
> | DiffIR (ICCV'23) | 26.94M |   120.99G   | 33.20 | 0.963 |
> | HI-Diff (ours)   | 28.49M |   142.62G   | 33.33 | 0.964 |
> | HI-Diff-2 (ours) | 23.99M |   125.47G   | 33.28 | 0.964 |
>
> 1. Compared with the DvSR, our methods have much smaller FLOPs because DM is performed in latent space. Meanwhile, our methods achieve better performance.
> 2. Compared with DiffIR, our methods perform better with comparable Params and FLOPs. This is because HI-Diff applies hierarchical integration, which is more suitable for processing non-uniform blur.
>
> **We provide more comprehensive comparisons in Tab. 1 of the PDF.**
>
>
>
> `Q1-3:` The paper does not analyze the effectiveness of the latent compression and evaluate the hyperparameters. It would be helpful to provide some ablation studies or analysis on the latent compression encoder.
>
> `A1-3:` Thanks for the suggestion. We ablation on the token number $N$ of latent space to analyze the effectiveness of latent compression. We set $N$ as 4, 16, and 64.
>
> All experiment settings are consistent with the ablation study (**Sec. 4.2**). Models are tested on GoPro, and the input size is 3×256×256 to calculate FLOPs.
>
> | $N$  | Params (M) | FLOPs (G) | PSNR (dB) |  SSIM  |
> | :--: | :--------: | :-------: | :-------: | :----: |
> |  4   |   23.985   |  125.05   |   32.12   | 0.9542 |
> |  16  |   23.986   |  125.47   |   32.24   | 0.9558 |
> |  64  |   23.990   |  127.33   |   32.28   | 0.9560 |
>
> With the increase of $N$, the performance increases, but the gain magnitude decreases. It may be because as $N$ increases, tokens become more redundant, thus the performance increase is limited. And the increase of redundant tokens also increases complexity and resource consumption. To balance performance, redundancy, and consumption, we choose $N$=16.
>
>
>
> `Q1-4:` Lack of one-stage ablation study or a clearer explanation is needed.
>
> `A1-4:` Thanks for pointing it out. We explain it below.
>
> 1. In our two-stage training architecture, the first stage uses the **ground truth image** as input to participate in model training. However, the second stage only adopts the **blurry image** as input. Therefore, we cannot combine the two stages into one stage. And we do not conduct the one-stage ablation study.
> 2. As we mentioned in **Sec 3.2** (Limitations) of the **supplementary material**, two-stage training is more tedious than one-stage training, which is a shortcoming of our method. One of our follow-up research directions is to explore how to conduct efficient one-stage training.
>
>
>
> `Q1-5:` Can you please elaborate on the structure and parameters of the diffusion model in more detail?
>
> `A1-5:` Thanks for asking for those details. We clarify them as follows.
>
> **For the structure**: The main component of the diffusion model is the Denoising Network (DN), which consists of some Linear Layers and MLP-Mixer Layers. The structure of DN is as follows.
>
> ```shell
> input -> (Linear+LRelU) -> (MLP-Mixer)×4 -> (Linear+LRelU) -> output
> ```
>
> Meanwhile, we apply the same diffusion and reverse process as DDPM (scheduler) [14].
>
> **For parameters**: We provide the Params and FLOPs of the diffusion model. We separately calculate FLOPs (input size: 16×256) for one diffusion step and total steps.
>
> | Params | FLOPs (one step) | FLOPs (total steps) | Step |
> | :----: | :--------------: | :-----------------: | :--: |
> | 2.63M  |      42.37M      |       338.95M       |  8   |
>
>
>
> `Q1-6:` It would be beneficial to include some visual results showcasing the effects of the diffusion process, if available.
>
> `A1-6:` Thanks for your valuable suggestion. We provide some visual results in **Fig. 1 of the PDF**.
>
> The blurred image gradually becomes sharp as the diffusion (reverse) process proceeds.
>
> Meanwhile, when the prior is Gaussian noise (i.e., $\mathbf{z}_{8}$), the output of Transformer is **not noise**. This may be because Transformer features actively ignore the invalid priors through cross-attention when fused with priors.
>
>
>
> `Q1-7:` In the event of replacing the diffusion model with alternative restoration models like transformer-based approaches, would the proposed method still maintain its effectiveness?
>
> `A1-7:` Thanks for your suggestions. We find that replacing DM with Transformer degrades model performance.
>
> **We have responded to another similar question, `Q1-1`. Please refer to `A1-1` for more details.**

---

> > ### Author Response · Authors · 2023-08-17
> > **Follow-up discussions with Reviewer R1 (mQcX)**
> >
> > Dear Reviewer mQcX,
> >
> > We thank you for your valuable review time and comments. We have responded to the related questions, which we believe have covered your concerns.
> >
> > 1. We provide analysis and experiments (comparing it with Transformer) to show the **superiority** of the **diffusion model** (DM).
> > 2. We compare our method with other DMs (e.g., **DvSR** and **DiffIR**) for image deblurring in terms of performance and complexity. Our method **outperforms** other DMs.
> > 3. We conduct an ablation study on **latent compression** (i.e., the token number N) to show its effectiveness.
> > 4. We analyze our shortcomings and follow-up research directions regarding **one-stage** training.
> > 5. We clarify the **structur**e and **parameters** of the DM.
> > 6. We provide **visualizations** to showcase the effects of the diffusion process.
> >
> > We hope to discuss further with you whether or not your concerns have been addressed. Please let us know if you still have any unsolved or other concerns. Then, we have enough time to provide further feedback. Thanks.
> >
> > Best,
> >
> > Authors

---

> > > ### Comment · Reviewer_mQcX · 2023-08-18
> > > **Thanks to the authors**
> > >
> > > I appreciate the effort in the provided rebuttal. Generally, I believe my concerns have been well addressed. The quality of the paper will improve by addressing these concerns in the revision. I have raised my rating accordingly.
> > >
> > > Furthermore, considering the paper's focus on realistic image deblurring, I recommend that the authors expand their qualitative experiments to include more real blurry datasets.

---

> > > > ### Author Response · Authors · 2023-08-18
> > > > **Thanks Reviewer mQcX for approving our work**
> > > >
> > > > Dear Reviewer mQcX,
> > > >
> > > > Thanks for agreeing that our response solves your concerns.
> > > >
> > > > For "expand their qualitative experiments to include more real blurry datasets". Thanks for your valuable suggestions. We will conduct qualitative experiments on more real blurry datasets (e.g., RWBI) in the revision.
> > > >
> > > > Best,
> > > >
> > > > Authors

---

> ### Comment · Area_Chair_HRVD · 2023-08-18
>
> Dear Reviewer mQcX: Your review states Soundness: 2 fair, Presentation: 2 fair, Contribution: 2 fair; yet your Rating is 7 Accept. This doesn't make much sense. Please respond and clarify.

---

> > ### Comment · Reviewer_mQcX · 2023-08-19
> >
> > Dear AC, the author's rebuttal has well addressed my concerns, and I forgot to update these relevant ratings. I have now updated the ratings.

---

### Author Rebuttal · Authors · 2023-08-08

## Response to all reviewers and area chairs for a brief summary

Dear reviewers and area chairs,

We thank all reviewers and area chairs for their valuable time and comments.

We are encouraged that:

1. Reviewer mQcX and Reviewer ymRm agree that our method is novel.
2. Reviewer ymRm thinks our experiments are extensive, and Reviewer w6Fg thinks the ablation study is complete.
3. All reviewers recognize that our method achieves state-of-the-art performance.



We have responded to each reviewer individually to address any comments. We would like to give a brief summary.

1. We explain the **reason for applying the diffusion model** (DM), and conduct additional experiments (comparing it with Transformer) to demonstrate **its superiority and importance**. Meanwhile, we also analyze the rationality of the loss for DM.
2. We provide comprehensive **comparisons with other DMs** for image deblurring in terms of performance and complexity.
3. We provide analyses and experiments to clarify the **differences** between our method and DiffIR, and show the **novelty** of our work.
4. We clarify the **details of our method**, including the structure and parameters of DM, differences from Restormer, and the settings of models on different datasets.
5. We analyze and experiment on **latent compression**. Meanwhile, we analyze our shortcomings and follow-up directions regarding **one-stage** training.
6. We provide the **latency comparisons** and analyze more on priors.
7. Finally, we provide more **visualizations** to enhance the solidity of the work, including the diffusion process and some failure cases.

We thank all reviewers and area chairs again!

Best,

Authors

---

### Decision · Program_Chairs · 2023-09-21

**Decision:**

Accept (spotlight)

**Comment:**

This is a nice paper and the reviewers, after having seen considered the rebuttal, seem to largely agree that it is a strong enough contribution to be included in the conference. I congratulate the authors.

I have a couple of suggestions for the References. The inclusion of the two works listed below will serve only a pedagogic purpose - neither work contains anything that takes away from the novelty of your contribution. Please include a brief mention of them in the related works section.

https://arxiv.org/abs/2308.09388
https://openreview.net/forum?id=VmyFF5lL3F